# CAQL: Continuous Action Q-Learning

**Moonkyung Ryu\*, Yinlam Chow\*, Ross Anderson, Christian Tjandraatmadja, Craig Boutilier**
Google Research
{mkryu,yinlamchow,rander,ctjandra,cboutilier}@google.com

## ABSTRACT

Value-based reinforcement learning (RL) methods like Q-learning have shown success in a variety of domains. One challenge in applying Q-learning to *continuous-action* RL problems, however, is the continuous action maximization (*max-Q*) required for optimal Bellman backup. In this work, we develop *CAQL*, a (class of) algorithm(s) for continuous-action Q-learning that can use several *plug-and-play* optimizers for the max-Q problem. Leveraging recent optimization results for deep neural networks, we show that max-Q can be solved optimally using *mixed-integer programming (MIP)*. When the Q-function representation has sufficient power, MIP-based optimization gives rise to better policies and is more robust than approximate methods (e.g., gradient ascent, cross-entropy search). We further develop several techniques to accelerate inference in CAQL, which despite their approximate nature, perform well. We compare CAQL with state-of-the-art RL algorithms on benchmark continuous-control problems that have different degrees of action constraints and show that CAQL outperforms policy-based methods in heavily constrained environments, often dramatically.

## 1 INTRODUCTION

Reinforcement learning (RL) has shown success in a variety of domains such as games (Mnih et al., 2013) and recommender systems (RSs) (Gauci et al., 2018). When the action space is finite, value-based algorithms such as Q-learning (Watkins & Dayan, 1992), which implicitly finds a policy by learning the optimal value function, are often very efficient because action optimization can be done by exhaustive enumeration. By contrast, in problems with a continuous action spaces (e.g., robotics (Peters & Schaal, 2006)), policy-based algorithms, such as policy gradient (PG) (Sutton et al., 2000; Silver et al., 2014) or cross-entropy policy search (CEPS) (Mannor et al., 2003; Kalashnikov et al., 2018), which directly learn a return-maximizing policy, have proven more practical. Recently, methods such as ensemble critic (Fujimoto et al., 2018) and entropy regularization (Haarnoja et al., 2018) have been developed to improve the performance of policy-based RL algorithms.

Policy-based approaches require a reasonable choice of policy parameterization. In some continuous control problems, Gaussian distributions over actions conditioned on some state representation is used. However, in applications such as RSs, where actions often take the form of high-dimensional item-feature vectors, policies cannot typically be modeled by common action distributions. Furthermore, the admissible action set in RL is constrained in practice, for example, when actions must lie within a specific range for safety (Chow et al., 2018). In RSs, the admissible actions are often *random functions* of the state (Boutilier et al., 2018). In such cases, it is non-trivial to define policy parameterizations that handle such factors. On the other hand, value-based algorithms are well-suited to these settings, providing potential advantage over policy methods. Moreover, at least with linear function approximation (Melo & Ribeiro, 2007), under reasonable assumptions, Q-learning converges to optimality, while such optimality guarantees for non-convex policy-based methods are generally limited (Fazel et al., 2018). Empirical results also suggest that value-based methods are more data-efficient and less sensitive to hyper-parameters (Quillen et al., 2018). Of course, with large action spaces, exhaustive action enumeration in value-based algorithms can be expensive—-one solution is to represent actions with continuous features (Dulac-Arnold et al., 2015).

The main challenge in applying value-based algorithms to continuous-action domains is selecting optimal actions (both at training and inference time). Previous work in this direction falls into three broad categories. The first solves the inner maximization of the (optimal) Bellman residual loss using global nonlinear optimizers, such as the cross-entropy method (CEM) for QT-Opt (Kalashnikov et al., 2018), gradient ascent (GA) for actor-expert (Lim et al., 2018), and action discretization (Uther & Veloso, 1998; Smart & Kaelbling, 2000; Lazaric et al., 2008). However, these approaches do not guarantee optimality. The second approach restricts the Q-function parameterization so that the optimization problem is tractable. For instance, one can discretize the state and action spaces and use a tabular Q-function representation. However, due to the curse of dimensionality, discretizations

must generally be coarse, often resulting in unstable control. Millán et al. (2002) circumvents this issue by averaging discrete actions weighted by their Q-values. Wire-fitting (Gaskett et al., 1999; III & Klopf, 1993) approximates Q-values piecewise-linearly over a discrete set of points, chosen to ensure the maximum action is one of the extreme points. The normalized advantage function (NAF) (Gu et al., 2016) constructs the state-action advantage function to be quadratic, hence analytically solvable. Parameterizing the Q-function with an input-convex neural network (Amos et al., 2017) ensures it is concave. These restricted functional forms, however, may degrade performance if the domain does not conform to the imposed structure. The third category replaces optimal Q-values with a "soft" counterpart (Haarnoja et al., 2018): an entropy regularizer ensures that both the optimal Q-function and policy have closed-form solutions. However, the sub-optimality gap of this soft policy scales with the interval and dimensionality of the action space (Neu et al., 2017).

Motivated by the shortcomings of prior approaches, we propose *Continuous Action Q-learning (CAQL)*, a Q-learning framework for continuous actions in which the Q-function is modeled by a *generic* feed-forward neural network.[1] Our contribution is three-fold. First, we develop the CAQL framework, which minimizes the Bellman residual in Q-learning using one of several "plug-and-play" action optimizers. We show that "max-Q" optimization, when the Q-function is approximated by a deep ReLU network, can be formulated as a mixed-integer program (MIP) that solves max-Q optimally. When the Q-function has sufficient representation power, MIP-based optimization induces better policies and is more robust than methods (e.g., CEM, GA) that approximate the max-Q solution. Second, to improve CAQL's practicality for larger-scale applications, we develop three speed-up techniques for computing max-Q values: (i) *dynamic tolerance*; (ii) *dual filtering*; and (iii) *clustering*. Third, we compare CAQL with several state-of-the-art RL algorithms on several benchmark problems with varying degrees of action constraints. Value-based CAQL is generally competitive, and outperforms policy-based methods in heavily constrained environments, sometimes significantly. We also study the effects of our speed-ups through ablation analysis.

## 2 PRELIMINARIES

We consider an infinite-horizon, discounted Markov decision process (Puterman, 2014) with states $X$, (continuous) action space $A$, reward function $R$, transition kernel $P$, initial state distribution $\beta$ and discount factor $\gamma \in [0, 1)$, all having the usual meaning. A (stationary, Markovian) policy $\pi$ specifies a distribution $\pi(\cdot|x)$ over actions to be taken at state $x$. Let $\Delta$ be the set of such policies. The *expected cumulative return* of $\pi \in \Delta$ is $J(\pi) := \mathbb{E}[\sum_{t=0}^{\infty} \gamma^t r_t \mid P, R, x_0 \sim \beta, \pi]$. An optimal policy $\pi^*$ satisfies $\pi^* \in \arg\max_{\pi \in \Delta} J(\pi)$. The Bellman operator $F[Q](x, a) = R(x, a) + \gamma \sum_{x' \in X} P(x'|x, a) \max_{a' \in A} Q(x', a')$ over state-action value function $Q$ has unique fixed point $Q^*(x, a)$ (Puterman, 2014), which is the optimal Q-function $Q^*(x, a) = \mathbb{E}[\sum_{t=0}^{\infty} \gamma^t R(x_t, a_t) \mid x_0 = x, a_0 = a, \pi^*]$. An optimal (deterministic) policy $\pi^*$ can be extracted from $Q^*$: $\pi^*(a|x) = \mathbf{1}\{a = a^*(x)\}$, where $a^*(x) \in \arg\max_a Q^*(x, a)$.

For large or continuous state/action spaces, the optimal Q-function can be approximated, e.g., using a deep neural network (DNN) as in DQN (Mnih et al., 2013). In DQN, the value function $Q_\theta$ is updated using the value label $r + \gamma \max_{a'} Q_{\theta^{\text{target}}}(x', a')$, where $Q_{\theta^{\text{target}}}$ is a *target* Q-function. Instead of training these weights jointly, $\theta^{\text{target}}$ is updated in a separate iterative fashion using the previous $\theta$ for a fixed number of training steps, or by averaging $\theta^{\text{target}} \leftarrow \tau\theta + (1 - \tau)\theta^{\text{target}}$ for some small momentum weight $\tau \in [0, 1]$ (Mnih et al., 2016). DQN is *off-policy*—the target is valid no matter how the experience was generated (as long as it is sufficiently exploratory). Typically, the loss is minimized over mini-batches $B$ of past data $(x, a, r, x')$ sampled from a large experience replay buffer $R$ (Lin & Mitchell, 1992). One common loss function for training $Q_{\theta^*}$ is *mean squared Bellman error*: $\min_\theta \sum_{i=1}^{|B|} (Q_\theta(x_i, a_i) - r_i - \gamma \max_{a'} Q_{\theta^{\text{target}}}(x'_i, a'))^2$. Under this loss, RL can be viewed as $\ell_2$-regression of $Q_\theta(\cdot, \cdot)$ w.r.t. target labels $r + \gamma \max_{a'} Q_{\theta^{\text{target}}}(x', a')$. We augment DQN, using *double Q-learning* for more stable training (Hasselt et al., 2016), whose loss is:

$$\min_\theta \sum_{i=1}^{|B|} \left( r_i + \gamma Q_{\theta^{\text{target}}}(x'_i, \arg\max_{a'} Q_\theta(x'_i, a')) - Q_\theta(x_i, a_i) \right)^2. \tag{1}$$

A *hinge loss* can also be used in Q-learning, and has connections to the linear programming (LP) formulation of the MDP (Puterman (2014)). The optimal $Q$-network weights can be specified as: $\min_\theta \frac{1}{|B|} \sum_{i=1}^{|B|} Q_\theta(x_i, a_i) + \lambda (r_i + \gamma \max_{a' \in A} Q_\theta(x'_i, a') - Q_\theta(x_i, a_i))_+$, where $\lambda > 0$ is a tun-

---

[1]Results can be extended to handle convolutional NNs, but are omitted for brevity.

able penalty w.r.t. constraint: $r + \gamma \max_{a' \in A} Q_\theta(x', a') \leq Q_\theta(x, a), \forall (x, a, r, x') \in B$. To stabilize training, we replace the $Q$-network of the inner maximization with the target $Q$-network and the optimal Q-value with the double-Q label, giving (see Appendix A for details):

$$\min_\theta \frac{1}{|B|} \sum_{i=1}^{|B|} Q_\theta(x_i, a_i) + \lambda \left( r_i + \gamma Q_{\theta^{\text{target}}}(x_i', \arg\max_{a'} Q_\theta(x_i', a')) - Q_\theta(x_i, a_i) \right)_+ . \quad (2)$$

In this work, we assume the $Q$-function approximation $Q_\theta$ to be a feed-forward network. Specifically, let $Q_\theta$ be a $K$-layer feed-forward NN with state-action input $(x, a)$ (where $a$ lies in a $d$-dimensional real vector space) and hidden layers arranged according to the equations:

$$z_1 = (x, a), \ \hat{z}_j = W_{j-1} z_{j-1} + b_{j-1}, \ z_j = h(\hat{z}_j), \ j = 2, \ldots, K, \ Q_\theta(x, a) := c^\top \hat{z}_K,^2 \quad (3)$$

where $(W_j, b_j)$ are the multiplicative and bias weights, $c$ is the output weight of the $Q$-network, $\theta = \left( c, \{(W_j, b_j)\}_{j=1}^{K-1} \right)$ are the weights of the $Q$-network, $\hat{z}_j$ denotes pre-activation values at layer $j$, and $h(\cdot)$ is the (component-wise) activation function. For simplicity, in the following analysis, we restrict our attention to the case when the activation functions are ReLU's. We also assume that the action space $A$ is a $d$-dimensional $\ell_\infty$-ball $B_\infty(\overline{a}, \Delta)$ with some radius $\Delta > 0$ and center $\overline{a}$. Therefore, at any arbitrary state $x \in X$ the max-Q problem can be re-written as $q_x^* := \max_{a \in A} Q_\theta(x, a) = \max_{\{\hat{z}_j\}_{j=2}^K, \{z_j\}_{j=2}^{K-1}} \left\{ c^\top \hat{z}_K : z_1 = (x, a), \ a \in B_\infty(\overline{a}, \Delta), \text{ eqs. (3)} \right\}$. While the above formulation is intuitive, the nonlinear equality constraints in the neural network formulation (3) makes this problem non-convex and NP-hard (Katz et al., 2017).

## 3 CONTINUOUS ACTION Q-LEARNING ALGORITHM

Policy-based methods (Silver et al., 2014; Fujimoto et al., 2018; Haarnoja et al., 2018) have been widely-used to handle continuous actions in RL. However, they suffer from several well-known difficulties, e.g., (i) modeling high-dimensional action distributions, (ii) handling action constraints, and (iii) data-inefficiency. Motivated by earlier work on value-based RL methods, such as QT-Opt (Kalashnikov et al., 2018) and actor-expert (Lim et al., 2018), we propose *Continuous Action Q-learning (CAQL)*, a general framework for continuous-action value-based RL, in which the Q-function is parameterized by a NN (Eq. 3). One novelty of CAQL is the formulation of the "max-Q" problem, i.e., the inner maximization in (1) and (2), as a mixed-integer programming (MIP).

The benefit of the MIP formulation is that it guarantees that we find the optimal action (and its true bootstrapped Q-value) when computing target labels (and at inference time). We show empirically that this can induce better performance, especially when the Q-network has sufficient representation power. Moreover, since MIP can readily model linear and combinatorial constraints, it offers considerable flexibility when incorporating complex action constraints in RL. That said, finding the optimal Q-label (e.g., with MIP) is computationally intensive. To alleviate this, we develop several approximation methods to systematically reduce the computational demands of the inner maximization. In Sec. 3.2, we introduce the *action function* to approximate the $\arg\max$-policy at inference time, and in Sec. 4 we propose three techniques, *dynamic tolerance*, *dual filtering*, and *clustering*, to speed up max-Q computation during training.

### 3.1 PLUG-N-PLAY MAX-Q OPTIMIZERS

In this section, we illustrate how the max-Q problem, with the Q-function represented by a ReLU network, can be formulated as a MIP, which can be solved using off-the-shelf optimization packages (e.g., SCIP (Gleixner et al., 2018), CPLEX (CPLEX, 2019), Gurobi (Gurobi, 2019)). In addition, we detail how approximate optimizers, specifically, gradient ascent (GA) and the cross-entropy method (CEM), can trade optimality for speed in max-Q computation within CAQL. For ease of exposition, we focus on Q-functions parameterized by a feedforward ReLU network. Extending our methodology (including the MIP formulation) to convolutional networks (with ReLU activation and max pooling) is straightforward (see Anderson et al. (2019)). While GA and CEM can handle generic activation functions beyond ReLU, our MIP requires additional approximations for those that are not piecewise linear.

**Mixed-Integer Programming (MIP)** A trained feed-forward ReLU network can be modeled as a MIP by formulating the nonlinear activation function at each neuron with binary constraints. Specifically, for a ReLU with pre-activation function of form $z = \max\{0, w^\top x + b\}$, where $x \in [\ell, u]$ is a

---

$^2$Without loss of generality, we simplify the NN by omitting the output bias and output activation function.

$d$-dimensional bounded input, $w \in \mathbb{R}^d$, $b \in \mathbb{R}$, and $\ell, u \in \mathbb{R}^d$ are the weights, bias and lower-upper bounds respectively, consider the following set with a binary variable $\zeta$ indicating whether the ReLU is active or not:

$$R(w, b, \ell, u) = \left\{ (x, z, \zeta) \ \middle| \ \begin{array}{l} z \geq w^\top x + b, \ z \geq 0, \ z \leq w^\top x + b - M^-(1 - \zeta), \ z \leq M^+ \zeta, \\ (x, z, \zeta) \in [\ell, u] \times \mathbb{R} \times \{0, 1\} \end{array} \right\}.$$

In this formulation, both $M^+ = \max_{x \in [\ell, u]} w^\top x + b$ and $M^- = \min_{x \in [\ell, u]} w^\top x + b$ can be computed in linear time in $d$. We assume $M^+ > 0$ and $M^- < 0$, otherwise the function can be replaced by $z = 0$ or $z = w^\top x + b$. These constraints ensure that $z$ is the output of the ReLU: If $\zeta = 0$, then they are reduced to $z = 0 \geq w^\top x + b$, and if $\zeta = 1$, then they become $z = w^\top x + b \geq 0$.

This can be extended to the ReLU network in (3) by *chaining* copies of intermediate ReLU formulations. More precisely, if the ReLU Q-network has $m_j$ neurons in layer $j \in \{2, \ldots, K\}$, for any given state $x \in X$, the max-Q problem can be reformulated as the following MIP:

$$q_x^* = \max \quad c^\top z_K \tag{4}$$
$$\text{s.t.} \quad z_1 := a \in B_\infty(\bar{a}, \Delta),$$
$$(z_{j-1}, z_{j,i}, \zeta_{j,i}) \in R(W_{j,i}, b_{j,i}, \ell_{j-1}, u_{j-1}), \ j \in \{2, \ldots, K\}, \ i \in \{1, \ldots, m_j\},$$

where $\ell_1 = \bar{a} - \Delta, u_1 = \bar{a} + \Delta$ are the (action) input-bound vectors. Since the output layer of the ReLU NN is linear, the MIP objective is linear as well. Here, $W_{j,i} \in \mathbb{R}^{m_j}$ and $b_{j,i} \in \mathbb{R}$ are the weights and bias of neuron $i$ in layer $j$. Furthermore, $\ell_j, u_j$ are interval bounds for the outputs of the neurons in layer $j$ for $j \geq 2$, and computing them can be done via interval arithmetic or other propagation methods (Weng et al., 2018) from the initial action space bounds (see Appendix C for details). As detailed by Anderson et al. (2019), this can be further tightened with additional constraints, and its implementation can be found in the *tf.opt* package described therein. As long as these bounds are redundant, having these additional box constraints will not affect optimality. We emphasize that the MIP returns *provably global optima*, unlike GA and CEM. Even when interrupted with stopping conditions such as a time limit, MIP often produces high-quality solutions in practice.

In theory, this MIP formulation can be solved in time exponential on the number of ReLUs and polynomial on the input size (e.g., by naively solving an LP for each binary variable assignment). In practice however, a modern MIP solver combines many different techniques to significantly speed up this process, such as branch-and-bound, cutting planes, preprocessing techniques, and primal heuristics (Linderoth & Savelsbergh, 1999). Versions of this MIP model have been used in neural network verification (Cheng et al., 2017; Lomuscio & Maganti, 2017; Bunel et al., 2018; Dutta et al., 2018; Fischetti & Jo, 2018; Anderson et al., 2019; Tjeng et al., 2019) and analysis (Serra et al., 2018; Kumar et al., 2019), but its *application to RL is novel*. While Say et al. (2017) also proposed a MIP formulation to solve the planning problem with non-linear state transition dynamics model learned with a NN, it is different than ours, which solves the max-Q problem.

**Gradient Ascent** GA (Nocedal & Wright, 2006) is a simple first-order optimization method for finding the (local) optimum of a differentiable objective function, such as a neural network Q-function. At any state $x \in X$, given a "seed" action $a_0$, the optimal action $\arg\max_a Q_\theta(x, a)$ is computed iteratively by $a_{t+1} \leftarrow a_t + \eta \nabla_a Q_\theta(x, a)$, where $\eta > 0$ is a step size (either a tunable parameter or computed using back-tracking line search (Nocedal & Yuan, 1998)). This process repeats until convergence, $|Q_\theta(x, a_{t+1}) - Q_\theta(x, a_t)| < \epsilon$, or a maximum iteration count is reached.

**Cross-Entropy Method** CEM (Rubinstein, 1999) is a derivative-free optimization algorithm. At any given state $x \in X$, it samples a batch of $N$ actions $\{a_i\}_{i=1}^N$ from $A$ using a fixed distribution (e.g., a Gaussian) and ranks the corresponding Q-values $\{Q_\theta(x, a_i)\}_{i=1}^N$. Using the top $K < N$ actions, it then updates the sampling distribution, e.g., using the sample mean and covariance to update the Gaussian. This is repeated until convergence or a maximum iteration count is reached.

### 3.2 ACTION FUNCTION

In traditional Q-learning, the policy $\pi^*$ is "implemented" by acting greedily w.r.t. the learned Q-function: $\pi^*(x) = \arg\max_a Q_\theta(x, a)$.[3] However, computing the optimal action can be expensive in the continuous case, which may be especially problematic at inference time (e.g., when computational power is limited in, say embedded systems, or real-time response is critical). To mitigate the problem, we can use an *action function* $\pi_w : X \to A$—effectively a trainable actor network—to

---

[3]Some exploration strategy may be incorporated as well.

approximate the greedy-action mapping $\pi^*$. We train $\pi_w$ using training data $B = \{(x_i, q_i^*)\}_{i=1}^{|B|}$, where $q_i^*$ is the max-Q label at state $x_i$. Action function learning is then simply a supervised regression problem: $w^* \in \arg\min_w \sum_{i=1}^{|B|} (q_i^* - Q_\theta(x_i, \pi_w(x_i)))^2$. This is similar to the notion of "distilling" an optimal policy from max-Q labels, as in actor-expert (Lim et al., 2018). Unlike actor-expert—a separate stochastic policy network is jointly learned with the Q-function to maximize the likelihood with the underlying optimal policy—our method learns a state-action mapping to approximate $\arg\max_a Q_\theta(x, a)$—this does not require distribution matching and is generally more stable. The use of action function in CAQL is simply *optional* to accelerate data collection and inference.

## 4  ACCELERATING MAX-Q COMPUTATION

In this section, we propose three methods to speed up the computationally-expensive max-Q solution during training: (i) dynamic tolerance, (ii) dual filtering, and (iii) clustering.

**Dynamic Tolerance**  Tolerance plays a critical role in the stopping condition of nonlinear optimizers. Intuitively, in the early phase of CAQL, when the Q-function estimate has high Bellman error, it may be wasteful to compute a highly accurate max-Q label when a crude estimate can already guide the gradient of CAQL to minimize the Bellman residual. We can speed up the max-Q solver by dynamically adjusting its tolerance $\tau > 0$ based on (a) the *TD-error*, which measures the estimation error of the optimal Q-function, and (b) the *training step* $t > 0$, which ensures the bias of the gradient (induced by the sub-optimality of max-Q solver) vanishes asymptotically so that CAQL converges to a stationary point. While relating tolerance with the Bellman residual is intuitive, it is impossible to calculate that without knowing the max-Q label. To resolve this circular dependency, notice that the action function $\pi_w$ approximates the optimal policy, i.e., $\pi_w(\cdot|x) \approx \arg\max_a Q_\theta(x, \cdot)$. We therefore replace the optimal policy with the action function in Bellman residual and propose the dynamic tolerance: $\tau_t := \frac{1}{|B|} \sum_{i=1}^{|B|} |r_i + \gamma Q_{\theta_t^{\text{target}}}(x_i', \pi_{w_t}(x_i')) - Q_{\theta_t}(x_i, a_i)| \cdot k_1 \cdot k_2^t$, where $k_1 > 0$ and $k_2 \in [0, 1)$ are tunable parameters. Under standard assumptions, CAQL with dynamic tolerance $\{\tau_t\}$ converges a.s. to a stationary point (Thm. 1, (Carden, 2014)).

**Dual Filtering**  The main motivation of *dual filtering* is to reduce the number of max-Q problems at each CAQL training step. For illustration, consider the formulation of hinge Q-learning in (2). Denote by $q_{x', \theta^{\text{target}}}^*$ the max-Q label w.r.t. the target Q-network and next state $x'$. The structure of the hinge penalty means the TD-error corresponding to sample $(x, a, x', r)$ is *inactive* whenever $q_{x', \theta^{\text{target}}}^* \leq (Q_\theta(x, a) - r)/\gamma$—this data can be discarded. In dual filtering, we efficiently estimate an upper bound on $q_{x', \theta^{\text{target}}}^*$ using some convex relaxation to determine which data can be discarded before max-Q optimization. Specifically, recall that the main source of non-convexity in (3) comes from the equality constraint of the ReLU activation function at each NN layer. Similar to MIP formulation, assume we have component-wise bounds $(l_j, u_j)$, $j = 2, \ldots, K - 1$ on the neurons, such that $l_j \leq \hat{z}_j \leq u_j$. The ReLU equality constraint $\mathcal{H}(l, u) := \{(h, k) \in \mathbb{R}^2 : h \in [l, u], k = [h]_+\}$ can be relaxed using a convex *outer-approximation* (Wong & Kolter, 2017): $\tilde{\mathcal{H}}(l, u) := \{(h, k) \in \mathbb{R}^2 : k \geq h, k \geq 0, -uh + (u - l)k \leq -ul\}$. We use this approximation to define the *relaxed* NN equations, which replace the nonlinear equality constraints in (3) with the convex set $\tilde{\mathcal{H}}(l, u)$. We denote the optimal Q-value w.r.t. the relaxed NN as $\tilde{q}_{x'}^*$, which is by definition an upper bound on $q_{x'}^*$. Hence, the condition: $\tilde{q}_{x_i', \theta^{\text{target}}}^* \leq (Q_\theta(x, a) - r)/\gamma$ is a conservative certificate for checking whether the data $(x, a, x', r)$ is inactive. For further speed up, we estimate $\tilde{q}_{x'}^*$ with its dual upper bound (see Appendix C for derivations) $\tilde{q}_{x'} := -(\hat{\nu}_1)^\top \bar{a} - m \cdot \|\hat{\nu}_1\|_q - (\hat{\nu}_1)^\top x' + \sum_{j=2}^{K-1} \sum_{s \in \mathcal{I}_j} l_j(s)[\nu_j(s)]_+ - \sum_{j=1}^{K-1} \nu_{j+1}^\top b_i$, where $\nu$ is defined by the following recursion "dual" network: $\nu_K := -c$, $\hat{\nu}_j := W_j^\top \nu_{j+1}$, $j = 1, \ldots, K - 1$, $\nu_j := D_j \hat{\nu}_j$, $j = 2, \ldots, K - 1$, and $D_s$ is a diagonal matrix with $[D_j](s, s) = \mathbf{1}\{s \in \mathcal{I}_j^+\} + u_j(s)/(u_j(s) - l_j(s)) \cdot \mathbf{1}\{s \in \mathcal{I}_j\}$, and replace the above certificate with an even more conservative one: $\tilde{q}_{x', \theta^{\text{target}}} \leq (Q_\theta(x, a) - r)/\gamma$.

Although dual filtering is derived for hinge Q-learning, it also applies to the $\ell_2$-loss counterpart by replacing the optimal value $q_{x', \theta^{\text{target}}}^*$ with its dual upper-bound estimate $\tilde{q}_{x', \theta^{\text{target}}}$ whenever the verification condition holds (i.e., the TD error is negative). Since the dual estimate is greater than the primal, the modified loss function will be a lower bound of the original in (1), i.e., $(r + \gamma \tilde{q}_{x', \theta^{\text{target}}} - Q_\theta(x, a))^2 \leq (r + \gamma q_{x', \theta^{\text{target}}}^* - Q_\theta(x, a))^2$ whenever $r + \gamma \tilde{q}_{x', \theta^{\text{target}}} - Q_\theta(x, a) \leq 0$, which can stabilize training by *reducing over-estimation error*.

One can utilize the inactive samples in the action function ($\pi_w$) learning problem by replacing the max-Q label $q^*_{x',\theta}$ with its dual approximation $\widetilde{q}_{x',\theta}$. Since $\widetilde{q}_{x',\theta} \geq q^*_{x',\theta}$, this replacement will not affect optimality. [4]

**Clustering** To reduce the number of max-Q solves further still, we apply online state aggregation (Meyerson, 2001), which picks a number of centroids from the batch of next states $B'$ as the centers of $p$-metric balls with radius $b > 0$, such that the union of these balls form a minimum covering of $B'$. Specifically, at training step $t \in \{0, 1, \ldots\}$, denote by $C_t(b) \subseteq B'$ the set of next-state centroids. For each next state $c' \in C_t(b)$, we compute the max-Q value $q^*_{c',\theta^{\text{target}}} = \max_{a'} Q_{\theta^{\text{target}}}(c', a')$, where $a^*_{c'}$ is the corresponding optimal action. For all remaining next states $x' \in B' \setminus C_t(b)$, we approximate their max-Q values via first-order Taylor series expansion $\hat{q}_{x',\theta^{\text{target}}} := q^*_{c',\theta^{\text{target}}} + \langle \nabla_{x'} Q_{\theta^{\text{target}}}(x', a')|_{x'=c',a'=a^*_{c'}}, (x' - c') \rangle$ in which $c'$ is the closest centroid to $x'$, i.e., $c' \in \arg\min_{c' \in C_t(b)} \|x' - c'\|_p$. By the envelope theorem for arbitrary choice sets (Milgrom & Segal, 2002), the gradient $\nabla_{x'} \max_{a'} Q_{\theta^{\text{target}}}(x', a')$ is equal to $\nabla_{x'} Q_{\theta^{\text{target}}}(x', a')|_{a'=a^*_{x'}}$. In this approach the cluster radius $r > 0$ controls the number of max-Q computations, which trades complexity for accuracy in Bellman residual estimation. This parameter can either be a tuned or adjusted dynamically (similar to dynamic tolerance), e.g., $r_t = k_3 \cdot k_4^t$ with hyperparameters $k_3 > 0$ and $k_4 \in [0, 1)$. Analogously, with this exponentially-decaying cluster radius schedule we can argue that the bias of CAQL gradient (induced by max-Q estimation error due to clustering) vanishes asymptotically, and the corresponding Q-function converges to a stationary point. To combine clustering with dual filtering, we define $B'_{\text{df}}$ as the batch of next states that are *inconclusive* after dual filtering, i.e., $B'_{\text{df}} = \{x' \in B' : \widetilde{q}_{x',\theta^{\text{target}}} > (Q_\theta(x, a) - r)/\gamma\}$. Then instead of applying clustering to $B'$ we apply this method onto the refined batch $B'_{\text{df}}$.

Dynamic tolerance not only speeds up training, but also improves CAQL's performance (see Tables 4 and 5); thus, we recommend using it by default. Dual filtering and clustering both trade off training speed with performance. These are practical options—with tunable parameters—that allow practitioners to explore their utility in specific domains.

## 5 EXPERIMENTS ON MUJOCO BENCHMARKS

To illustrate the effectiveness of CAQL, we (i) compare several CAQL variants with several state-of-the-art RL methods on multiple domains, and (ii) assess the trade-off between max-Q computation speed and policy quality via ablation analysis.

**Comparison with Baseline RL Algorithms** We compare CAQL with four baseline methods, DDPG (Silver et al., 2014), TD3 (Fujimoto et al., 2018), and SAC (Haarnoja et al., 2018)—three popular policy-based deep RL algorithms—and NAF (Gu et al., 2016), a value-based method using an action-quadratic Q-function. We train CAQL using three different max-Q optimizers, MIP, GA, and CEM. Note that CAQL-CEM counterpart is similar to QT-Opt (Kalashnikov et al., 2018) and CAQL-GA reflects some aspects actor-expert (Lim et al., 2018). These CAQL variants allow assessment of the degree to which policy quality is impacted by Q-learning with optimal Bellman residual (using MIP) rather than an approximation (using GA or CEM), at the cost of steeper computation. To match the implementations of the baselines, we use $\ell_2$ loss when training CAQL. Further ablation analysis on CAQL with $\ell_2$ loss vs. hinge loss is provided in Appendix E.

We evaluate CAQL on one classical control benchmark (Pendulum) and five MuJoCo benchmarks (Hopper, Walker2D, HalfCheetah, Ant, Humanoid).[5] Different than most previous work, we evaluate the RL algorithms on domains not just with default action ranges, but also using smaller, *constrained* action ranges (see Table 6 in Appendix D for action ranges used in our experiments).[6] The motivation for this is two-fold: (i) To simulate real-world problems (Dulac-Arnold et al., 2019), where the restricted ranges represent the safe/constrained action sets; (ii) To validate the hypothesis that action-distribution learning in policy-based methods cannot easily handle such constraints, while CAQL does so, illustrating its flexibility. For problems with longer horizons, a larger neural network is often required to learn a good policy, which in turn significantly increases the complexity

---

[4]One can also use the upper bound $\widetilde{q}_{x',\theta}$ as the label for training the action function in Section 3.2. Empirically, this approach often leads to better policy performance.

[5]Since the objective of these experiments is largely to evaluate different RL algorithms, we do not exploit problem structures, e.g., symmetry, when training policies.

[6]Smaller action ranges often induce easier MIP problems in max-Q computation. However, given the complexity of MIP in more complex environments such as Walker2D, HalfCheetah, Ant, and Humanoid, we run experiments only with action ranges smaller than the defaults.

| Env. [Action range] | CAQL-MIP | CAQL-GA | CAQL-CEM | NAF | DDPG | TD3 | SAC |
|---|---|---|---|---|---|---|---|
| Pendulum [-0.66, 0.66] | **-339.5** ± 158.3 | -342.4 ± 151.6 | -394.6 ± 246.5 | -449.4 ± 280.5 | -407.3 ± 180.3 | -488.8 ± 232.3 | -789.6 ± 299.5 |
| Pendulum [-1, 1] | **-235.9** ± 122.7 | -237.0 ± 135.5 | -236.1 ± 116.3 | -312.7 ± 242.5 | -252.8 ± 163.0 | -279.5 ± 186.8 | -356.6 ± 288.7 |
| Pendulum [-2, 2] | -143.2 ± 161.0 | -145.5 ± 136.1 | -144.5 ± 208.8 | -145.2 ± 168.9 | -146.2 ± 257.6 | **-142.3** ± 195.9 | -163.3 ± 190.1 |
| Hopper [-0.25, 0.25] | **343.2** ± 62.6 | 329.7 ± 59.4 | 276.9 ± 97.4 | 237.8 ± 100.0 | 252.2 ± 98.1 | 217.1 ± 73.7 | 309.3 ± 73.0 |
| Hopper [-0.5, 0.5] | **411.7** ± 115.2 | 341.7 ± 139.9 | 342.9 ± 142.1 | 248.2 ± 113.2 | 294.5 ± 108.7 | 280.1 ± 80.0 | 309.1 ± 95.8 |
| Hopper [-1, 1] | **459.8** ± 144.9 | 427.5 ± 151.2 | 417.2 ± 145.4 | 245.9 ± 140.7 | 368.2 ± 139.3 | 396.3 ± 132.8 | 372.3 ± 138.5 |
| Walker2D [-0.25, 0.25] | 276.3 ± 118.5 | **285.6** ± 97.6 | 283.7 ± 104.6 | 219.9 ± 120.8 | 270.4 ± 104.2 | 250.0 ± 78.3 | 284.0 ± 114.5 |
| Walker2D [-0.5, 0.5] | 288.9 ± 118.1 | 295.6 ± 113.9 | **304.7** ± 116.1 | 233.7 ± 99.4 | 259.0 ± 110.7 | 243.8 ± 116.4 | 287.0 ± 128.3 |
| HalfCheetah [-0.25, 0.25] | **394.8** ± 43.8 | 337.4 ± 60.0 | 339.1 ± 137.9 | 247.3 ± 96.0 | 330.7 ± 98.9 | 264.3 ± 142.2 | 325.9 ± 38.6 |
| HalfCheetah [-0.5, 0.5] | 718.6 ± 199.9 | **736.4** ± 122.8 | 686.7 ± 224.1 | 405.1 ± 243.2 | 456.3 ± 238.5 | 213.8 ± 214.6 | 614.8 ± 69.4 |
| Ant [-0.1, 0.1] | 402.3 ± 27.4 | **406.2** ± 32.6 | 378.2 ± 39.7 | 295.0 ± 44.2 | 374.0 ± 35.9 | 268.9 ± 73.2 | 281.4 ± 65.3 |
| Ant [-0.25, 0.25] | 413.1 ± 60.0 | 443.1 ± 65.6 | 451.4 ± 54.8 | 323.0 ± 60.8 | 444.2 ± 63.3 | **472.3** ± 61.9 | 399.3 ± 59.2 |
| Humanoid [-0.1, 0.1] | 405.7 ± 112.5 | 431.9 ± 244.8 | 397.0 ± 145.7 | 392.7 ± 169.9 | **494.4** ± 182.0 | 352.8 ± 150.3 | 456.3 ± 112.4 |
| Humanoid [-0.25, 0.25] | 460.2 ± 143.2 | **622.8** ± 158.1 | 529.8 ± 179.9 | 374.6 ± 126.5 | 582.1 ± 176.7 | 348.1 ± 106.3 | 446.9 ± 103.9 |

Table 1: The mean ± standard deviation of (95-percentile) final returns with the best hyper-parameter configuration. CAQL significantly outperforms NAF on most benchmarks, as well as DDPG, TD3, and SAC on 11/14 benchmarks.

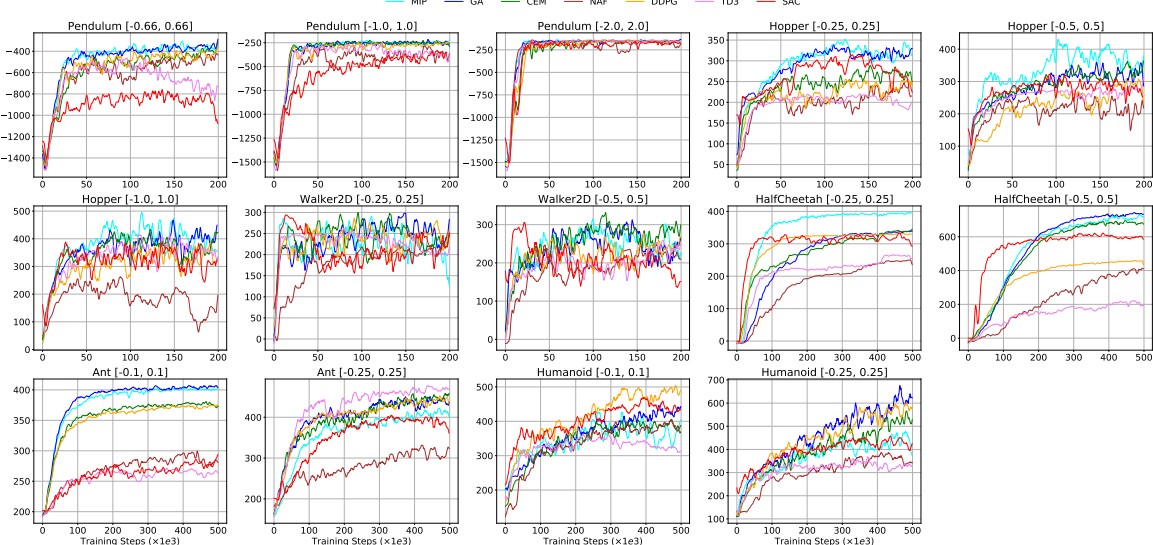

Figure 1: Mean cumulative reward of the best hyper parameter configuration over 10 random seeds. Data points are average over a sliding window of size 6. The length of an episode is limited to 200 steps. The training curves with standard deviation are given in Figure 4 in Appendix E.

of the MIP. To reduce this computational cost, we reduce episode length in each experiment from 1000 to 200 steps, and parameterize the Q-function with a relatively simple $32 \times 16$ feedforward ReLU network. With shorter episodes and smaller networks, the returns of our experiments are lower than those reported in state-of-the-art RL benchmarks (Duan et al., 2016). Details on network architectures and hyperparameters are described in Appendix D.

For the more difficult MuJoCo environments (i.e., Ant, HalfCheetah, Humanoid), the number of training steps is set to $500,000$, while for simpler ones (i.e., Pendulum, Hopper, Walker2D), it is set to $200,000$. Policy performance is evaluated every $1000$ training iterations, using a policy with no exploration. Each measurement is an average return over 10 episodes, each generated using a separate random seed. To smooth learning curves, data points are averaged over a sliding window of size 6. Similar to the setting of Lim et al. (2018), CAQL measurements are based on trajectories that are generated by the learned action function instead of the optimal action w.r.t. the Q-function.

Table 1 and Figure 1 show the average return of CAQL and the baselines under the best hyperparameter configurations. CAQL significantly outperforms NAF on most benchmarks, as well as DDPG, TD3, and SAC on 11 of 14 benchmarks. Of all the CAQL policies, those trained using MIP are among the best performers in low-dimensional benchmarks (e.g., Pendulum and Hopper). This verifies our conjecture about CAQL: Q-learning with optimal Bellman residual (using MIP) performs better than using approximation (using GA, CEM) when the Q-function has sufficient representation power (which is more likely in low-dimensional tasks). Moreover, CAQL-MIP policies have slightly lower variance than those trained with GA and CEM on most benchmarks. Table 2 and Figure 2 show summary statistics of the returns of CAQL and the baselines on all 320 configurations

| Env. [Action range] | CAQL-MIP | CAQL-GA | CAQL-CEM | NAF | DDPG | TD3 | SAC |
|---|---|---|---|---|---|---|---|
| Pendulum [-0.66, 0.66] | -780.5 ± 345.0 | **-766.6** ± 344.2 | -784.7 ± 349.3 | -775.3 ± 353.4 | -855.2 ± 331.2 | -942.1 ± 308.3 | -1144.8 ± 195.3 |
| Pendulum [-1, 1] | -508.1 ± 383.2 | -509.7 ± 383.5 | **-500.7** ± 382.5 | -529.5 ± 377.4 | -623.3 ± 395.2 | -730.3 ± 389.4 | -972.0 ± 345.5 |
| Pendulum [-2, 2] | **-237.3** ± 487.2 | -250.6 ± 508.1 | -249.7 ± 488.5 | -257.4 ± 370.3 | -262.0 ± 452.6 | -361.3 ± 473.2 | -639.5 ± 472.7 |
| Hopper [-0.25, 0.25] | **292.7** ± 93.3 | 210.8 ± 125.3 | 196.9 ± 130.1 | 176.6 ± 109.1 | 178.8 ± 126.6 | 140.5 ± 106.5 | 225.0 ± 84.9 |
| Hopper [-0.5, 0.5] | **332.2** ± 119.7 | 222.2 ± 138.5 | 228.1 ± 135.7 | 192.8 ± 101.6 | 218.3 ± 129.6 | 200.2 ± 100.7 | 243.6 ± 81.6 |
| Hopper [-1, 1] | **352.2** ± 141.3 | 251.5 ± 153.6 | 242.3 ± 153.8 | 201.9 ± 126.2 | 248.0 ± 148.3 | 248.2 ± 124.4 | 263.6 ± 118.9 |
| Walker2D [-0.25, 0.25] | **247.6** ± 109.0 | 213.5 ± 111.3 | 206.7 ± 112.9 | 190.5 ± 117.5 | 209.9 ± 103.6 | 204.8 ± 113.3 | 224.5 ± 105.1 |
| Walker2D [-0.5, 0.5] | 213.1 ± 120.0 | 209.5 ± 112.5 | 209.3 ± 112.5 | 179.7 ± 100.9 | 210.8 ± 108.3 | 173.1 ± 101.1 | **220.9** ± 114.8 |
| HalfCheetah [-0.25, 0.25] | **340.9** ± 110.2 | 234.3 ± 136.5 | 240.4 ± 143.1 | 169.7 ± 123.7 | 228.9 ± 118.1 | 192.9 ± 136.8 | 260.7 ± 108.5 |
| HalfCheetah [-0.5, 0.5] | 395.1 ± 275.2 | **435.5** ± 273.7 | 377.5 ± 280.5 | 271.8 ± 226.9 | 273.8 ± 199.5 | 119.8 ± 139.6 | 378.3 ± 219.6 |
| Ant [-0.1, 0.1] | 319.4 ± 69.3 | **327.5** ± 67.5 | 295.5 ± 71.9 | 260.2 ± 53.1 | 298.4 ± 67.6 | 213.7 ± 40.8 | 205.9 ± 34.9 |
| Ant [-0.25, 0.25] | 362.3 ± 60.3 | 388.9 ± 63.9 | **392.9** ± 67.1 | 270.4 ± 72.5 | 381.9 ± 63.3 | 377.1 ± 93.5 | 314.3 ± 88.2 |
| Humanoid [-0.1, 0.1] | 326.6 ± 93.5 | 235.3 ± 165.4 | 227.7 ± 143.1 | 261.6 ± 154.1 | 259.0 ± 188.1 | 251.7 ± 127.5 | **377.5** ± 90.3 |
| Humanoid [-0.25, 0.25] | 267.0 ± 163.8 | 364.3 ± 215.9 | 309.4 ± 186.3 | 270.2 ± 124.6 | 347.3 ± 220.8 | 262.8 ± 109.2 | **381.0** ± 84.4 |

Table 2: The mean ± standard deviation of (95-percentile) final returns over all 320 configurations (32 hyper parameter combinations × 10 random seeds). CAQL policies are less sensitive to hyper parameters on 11/14 benchmarks.

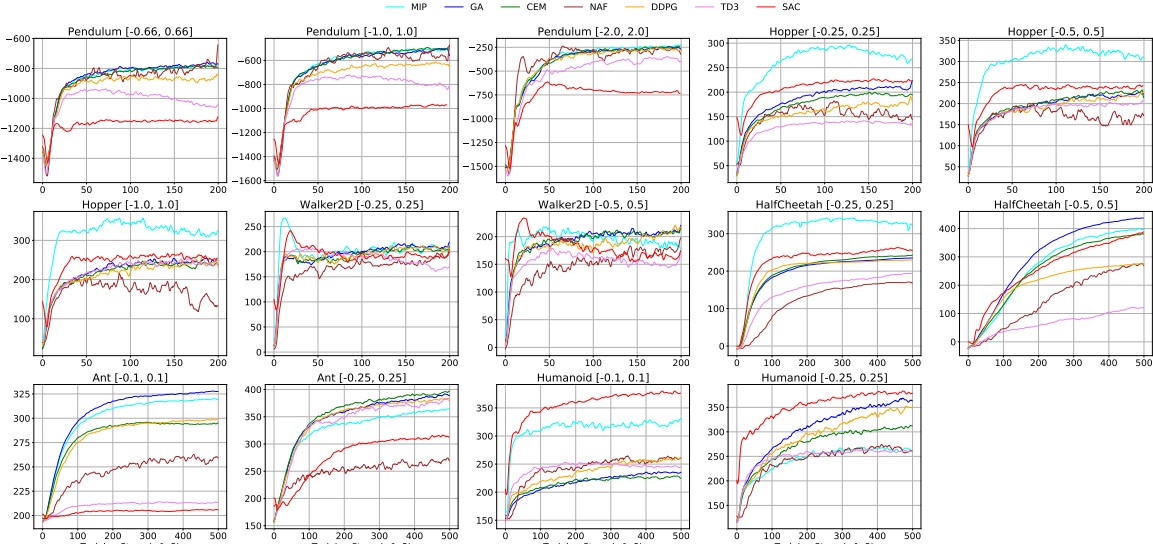

Figure 2: Mean cumulative reward over all 320 configurations (32 hyper parameter combinations × 10 random seeds). Data points are average over a sliding window of size 6. The length of an episode is limited to 200 steps. The training curves with standard deviation are in Figure 5 in Appendix E.

(32 hyperparameter combinations × 10 random seeds) and illustrates the sensitivity to hyperparameters of each method. CAQL is least sensitive in 11 of 14 tasks, and policies trained using MIP optimization, specifically, are best in 6 of 14 tasks. This corroborates the hypothesis that value-based methods are generally more robust to hyperparameters than their policy-based counterparts. Table 9 in Appendix E.1 compares the speed (in terms of average elapsed time) of various max-Q solvers (MIP, GA, and CEM), with MIP clearly the most computationally intensive.

We note that CAQL-MIP suffers from performance degradation in several high-dimensional environments with large action ranges (e.g., Ant [-0.25, 0.25] and Humanoid [-0.25, 0.25]). In these experiments, its performance is even worse than that of CAQL-GA or CAQL-CEM. We speculate that this is due to the fact that the small ReLU NN ($32 \times 16$) doesn't have enough representation power to accurately model the Q-functions in more complex tasks, and therefore optimizing for the true max-Q value using an inaccurate function approximation impedes learning.

We also test CAQL using the standard MuJoCo 1000-step episode length, using gradient ascent as the optimizer, and a Q-function is parameterized with a $200 \times 100$ feedforward ReLU network for Hopper and with $400 \times 300$ for the rest benchmarks. CAQL-GA is trained using dynamic tolerance and an action function but without dual filtering or clustering. Figure 6 in Appendix E shows that CAQL-GA performs better than, or similar to, the best of the baseline methods, except on Hopper [-0.25, 0.25]—SAC performed best in that setting, however, it suffers from very high performance variance.

**Ablation Analysis** We now study the effects of using dynamic tolerance, dual filtering, and clustering on CAQL via two ablation analyses. For simplicity, we experiment on standard benchmarks

| Env. [Action range] | GA | GA + DF | GA + DF + C(0.25) | GA + DF + C(0.5) | Dual |
|---|---|---|---|---|---|
| Pendulum [-2, 2] | **-144.6** ± 154.2 (R: 0.00%) | -146.1 ± 229.8 (R: 26.5%) | -146.7 ± 216.8 (R: 79.7%) | -149.9 ± 215.0 (R: 81.2%) | -175.1 ± 246.8 (R: 100%) |
| Hopper [-1, 1] | 414.9 ± 181.9 (R: 0.00%) | **424.8** ± 176.7 (R: 9.11%) | 396.3 ± 138.8 (R: 38.2%) | 371.2 ± 171.7 (R: 61.5%) | 270.2 ± 147.6 (R: 100%) |
| Walker2D [-1, 1] | **267.6** ± 107.3 (R: 0.00%) | 236.1 ± 99.6 (R: 12.4%) | 249.2 ± 125.1 (R: 33.6%) | 235.6 ± 108.1 (R: 50.8%) | 201.5 ± 136.0 (R: 100%) |
| HalfCheetah [-1, 1] | **849.0** ± 108.9 (R: 0.00%) | 737.3 ± 170.2 (R: 5.51%) | 649.5 ± 146.7 (R: 15.4%) | 445.4 ± 207.8 (R: 49.1%) | 406.6 ± 206.4 (R: 100%) |
| Ant [-0.5, 0.5] | **370.0** ± 106.5 (R: 0.00%) | 275.1 ± 92.1 (R: 3.19%) | 271.5 ± 94.3 (R: 14.3%) | 214.0 ± 75.4 (R: 31.1%) | 161.1 ± 54.7 (R: 100%) |
| Humanoid [-0.4, 0.4] | **702.7** ± 162.9 (R: 0.00%) | 513.9 ± 146.0 (R: 13.8%) | 458.2 ± 120.4 (R: 52.1%) | 387.7 ± 117.4 (R: 80.8%) | 333.3 ± 117.9 (R: 100%) |

Table 3: Ablation analysis on CAQL-GA with dual filtering and clustering, where both the mean ± standard deviation of (95-percentile) final returns and the average %-max-Q-reduction (in parenthesis) are based on the best configuration. See Figure 7 in Appendix E for training curves.

| Env. [Action range] | GA + Tol(1e-6) | GA + Tol(100) | GA + DTol(100,1e-6) | GA + DTol(1,1e-6) | GA + DTol(0.1,1e-6) |
|---|---|---|---|---|---|
| Pendulum [-2, 2] | -144.5 ± 195.6 (# GA Itr: 200) | -158.1 ± 165.0 (# GA Itr: 1) | -144.1 ± 159.2 (# GA Itr: 45.4) | **-143.7** ± 229.5 (# GA Itr: 58.1) | -144.2 ± 157.6 (# GA Itr: 69.5) |
| Hopper [-1, 1] | 371.4 ± 199.9 (# GA Itr: 200) | 360.4 ± 158.9 (# GA Itr: 1) | 441.4 ± 141.3 (# GA Itr: 46.0) | **460.0** ± 127.8 (# GA Itr: 59.5) | 452.5 ± 137.0 (# GA Itr: 78.4) |
| Walker2D [-1, 1] | 273.6 ± 112.4 (# GA Itr: 200) | 281.6 ± 121.2 (# GA Itr: 1) | 282.0 ± 104.5 (# GA Itr: 47.4) | **309.0** ± 118.8 (# GA Itr: 59.8) | 292.7 ± 113.8 (# GA Itr: 71.2) |
| HalfCheetah [-1, 1] | 837.5 ± 130.6 (# GA Itr: 200) | 729.3 ± 313.8 (# GA Itr: 1) | **896.6** ± 145.7 (# GA Itr: 91.2) | 864.4 ± 123.1 (# GA Itr: 113.5) | 894.0 ± 159.9 (# GA Itr: 140.7) |
| Ant [-0.5, 0.5] | 373.1 ± 118.5 (# GA Itr: 200) | 420.7 ± 148.8 (∗) (# GA Itr: 1) | 364.4 ± 111.324 (# GA Itr: 86.3) | 388.2 ± 110.7 (# GA Itr: 109.5) | **429.3** ± 139.3 (# GA Itr: 139.7) |
| Humanoid [-0.4, 0.4] | 689.8 ± 193.9 (# GA Itr: 200) | 500.2 ± 214.9 (# GA Itr: 1) | **716.2** ± 191.4 (# GA Itr: 88.6) | 689.5 ± 191.6 (# GA Itr: 115.9) | 710.9 ± 188.4 (# GA Itr: 133.4) |

Table 4: Ablation analysis on CAQL-GA with dynamic tolerance, where both the mean ± standard deviation of (95-percentile) final returns and the average number of GA iterations (in parenthesis) are based on the best configuration. See Figure 9 in Appendix E for training curves. NOTE: In (∗) the performance significantly drops after hitting the peak, and learning curve does not converge.

(with full action ranges), and primarily test CAQL-GA using an $\ell_2$ loss. Default values on tolerance and maximum iteration are 1e-6 and 200, respectively.

Table 3 shows how reducing the number of max-Q problems using dual filtering and clustering affects performance of CAQL. Dual filtering (DF) manages to reduce the number of max-Q problems (from 3.2% to 26.5% across different benchmarks), while maintaining similar performance with the unfiltered CAQL-GA. On top of dual filtering we apply clustering (C) to the set of inconclusive next states $B'_{df}$, in which the degree of approximation is controlled by the cluster radius. With a small cluster radius (e.g., $b = 0.1$), clustering further reduces max-Q solves without significantly impacting training performance (and in some cases it actually improves performance), though further increasing the radius would significant degrade performance. To illustrate the full trade-off of max-Q reduction versus policy quality, we also include the *Dual* method, which eliminates all max-Q computation with the dual approximation. Table 4 shows how dynamic tolerance influences the quality of CAQL policies. Compared with the standard algorithm, with a large tolerance ($\tau = 100$) GA achieves a notable speed up (with only 1 step per max-Q optimization) in training but incurs a loss in performance. GA with dynamic tolerance atttains the best of both worlds—it significantly reduces inner-maximization steps (from 29.5% to 77.3% across different problems and initial $\tau$ settings), while achieving good performance.

Additionally, Table 5 shows the results of CAQL-MIP with dynamic tolerance (i.e., optimality gap). This method significantly reduces both median and variance of the MIP elapsed time, while having better performance. Dynamic tolerance eliminates the high latency in MIP observed in the early phase of training (see Figure 3).

| Env. [Action range] | MIP + Tol(1e-4) | MIP + DTol(1,1e-4) |
|---|---|---|
| HalfCheetah [-0.5, 0.5] | 718.6 ± 199.9 (Med($\kappa$): 263.5, SD($\kappa$): 88.269) | **764.5** ± 132.9 (Med($\kappa$): 118.5, SD($\kappa$): 75.616) |
| Ant [-0.1, 0.1] | 402.3 ± 27.4 (Med($\kappa$): 80.7, SD($\kappa$): 100.945) | **404.9** ± 27.7 (Med($\kappa$): 40.3, SD($\kappa$): 24.090) |
| Ant [-0.25, 0.25] | 413.1 ± 60.0 (Med($\kappa$): 87.6, SD($\kappa$): 160.921) | **424.9** ± 60.9 (Med($\kappa$): 62.0, SD($\kappa$): 27.646) |
| Humanoid [-0.1, 0.1] | 405.7 ± 112.5 (Med($\kappa$): 145.7, SD($\kappa$): 27.381) | **475.0** ± 173.4 (Med($\kappa$): 29.1, SD($\kappa$): 10.508) |
| Humanoid [-0.25, 0.25] | **460.2** ± 143.2 (Med($\kappa$): 71.2, SD($\kappa$): 45.763) | 410.1 ± 174.4 (Med($\kappa$): 39.7, SD($\kappa$): 11.088) |

Table 5: Ablation analysis on CAQL-MIP with dynamic tolerance, where both the mean ± standard deviation of (95-percentile) final returns and the (median, standard deviation) of the elapsed time $\kappa$ (in msec) are based on the best configuration. See Figure 11 in Appendix E for training curves.

## 6 CONCLUSIONS AND FUTURE WORK

We proposed *Continuous Action Q-learning (CAQL)*, a general framework for handling continuous actions in value-based RL, in which the Q-function is parameterized by a neural network. While generic nonlinear optimizers can be naturally integrated with CAQL, we illustrated how the inner maximization of Q-learning can be formulated as mixed-integer programming when the Q-function is parameterized with a ReLU network. CAQL (with action function learning) is a general Q-learning framework that includes many existing value-based methods such as QT-Opt and actor-expert. Using several benchmarks with varying degrees of action constraint, we showed that the policy learned by CAQL-MIP generally outperforms those learned by CAQL-GA and CAQL-CEM; and CAQL is competitive with several state-of-the-art policy-based RL algorithms, and often outperforms them (and is more robust) in heavily-constrained environments. Future work includes: extending CAQL to the full batch learning setting, in which the optimal Q-function is trained using only offline data; speeding up the MIP computation of the max-Q problem to make CAQL more scalable; and applying CAQL to real-world RL problems.

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

## A  HINGE Q-LEARNING

Consider an MDP with states $X$, actions $A$, transition probability function $P$, discount factor $\gamma \in [0, 1)$, reward function $R$, and initial state distribution $\beta$. We want to find an optimal $Q$-function by solving the following optimization problem:

$$\min_Q \sum_{x \in X, a \in A} p(x, a) Q(x, a)$$

$$Q(x, a) \geq R(x, a) + \gamma \sum_{x' \in X} P(x'|x, a) \max_{a' \in A} Q(x', a'), \ \forall x \in X, a \in A. \tag{5}$$

The formulation is based on the LP formulation of MDP (see Puterman (2014) for more details). Here the distribution $p(x, a)$ is given by the data-generating distribution of the replay buffer $B$. (We assume that the replay buffer is large enough such that it consists of experience from almost all state-action pairs.) It is well-known that one can transform the above constrained optimization problem into an unconstrained one by applying a penalty-based approach (to the constraints). For simplicity, here we stick with a single constant penalty parameter $\lambda \geq 0$ (instead of going for a state-action Lagrange multiplier and maximizing that), and a hinge penalty function $(\cdot)_+$. With a given penalty hyper-parameter $\lambda \geq 0$ (that can be separately optimized), we propose finding the optimal $Q$-function by solving the following optimization problem:

$$\min_Q \sum_{x \in X, a \in A} p(x, a) Q(x, a) + \lambda \left( R(x, a) + \gamma \sum_{x' \in X} P(x'|x, a) \max_{a' \in A} Q(x', a') - Q(x, a) \right)_+. \tag{6}$$

Furthermore, recall that in many off-policy and offline RL algorithms (such as DQN), samples in form of $\{(x_i, a_i, r_i, x_i')\}_{i=1}^{|B|}$ are independently drawn from the replay buffer, and instead of the optimizing the original objective function, one goes for its unbiased sample average approximation (SAA). However, viewing from the objective function of problem (6), finding an unbiased SAA for this problem might be challenging, due to the non-linearity of hinge penalty function $(\cdot)_+$. Therefore, alternatively we turn to study the following unconstrained optimization problem:

$$\min_Q \sum_{x \in X, a \in A} p(x, a) Q(x, a) + \lambda \sum_{x' \in X} P(x'|x, a) \left( R(x, a) + \gamma \max_{a' \in A} Q(x', a') - Q(x, a) \right)_+. \tag{7}$$

Using the Jensen's inequality for convex functions, one can see that the objective function in (7) is an upper-bound of that in (6). Equality of the Jensen's inequality will hold in the case when transition function is deterministic. (This is similar to the argument of PCL algorithm.) Using Jensen's inequality one justifies that optimization problem (7) is indeed an eligible upper-bound optimization to problem (6).

Recall that $p(x, a)$ is the data-generation distribution of the replay buffer $B$. The unbiased SAA of problem (7) is therefore given by

$$\min_Q \frac{1}{N} \sum_{s=1}^N Q(x_i, a_i) + \lambda \left( r_i + \gamma \max_{a' \in A} Q(x_i', a') - Q(x_i, a_i) \right)_+, \tag{8}$$

where $\{(x_i, a_i, r_i, x_i')\}_{s=1}^N$ are the $N$ samples drawn independently from the replay buffer. In the following, we will find the optimal $Q$ function by solving this SAA problem. In general when the state and action spaces are large/uncountable, instead of solving the $Q$-function exactly (as in the tabular case), we turn to approximate the $Q$-function with its parametrized form $Q_\theta$, and optimize the set of real weights $\theta$ (instead of $Q$) in problem (8).

# B  CONTINUOUS ACTION Q-LEARNING ALGORITHM

---

**Algorithm 1** Continuous Action Q-learning (CAQL)

---
1: Define the maximum training epochs $T$, episode length $L$, and training steps (per epoch) $S$
2: Initialize the Q-function parameters $\theta$, target Q-function parameters $\theta^{\text{target}}$, and action function parameters $w$
3: Choose a max-Q solver OPT $\in \{\text{MIP}, \text{CEM}, \text{GA}\}$ (Section 3)
4: Select options for max-Q speed up (Section 4) using the following Boolean variables: (i) DTol: Dynamic Tolerance, (ii) DF: Dual Filtering, (iii) C: Clustering; Denote by OPT(Dtol) the final max-Q solver, constructed by the base solver and dynamic tolerance update rule (if used)
5: Initialize replay buffer $R$ of states, actions, next states and rewards
6: **for** $t \leftarrow 1, \dots, T$ **do**
7:     Sample an initial state $x_0$ from the initial distribution
8:     **for** $\ell \leftarrow 0, \dots, L-1$ **do** ▷ Online Data Collection
9:         Select action $a_\ell = clip(\pi_w(x_\ell) + \mathcal{N}(0, \sigma), l, u)$
10:         Execute action $a_\ell$ and observe reward $r_\ell$ and new state $x_{\ell+1}$
11:         Store transition $(x_\ell, a_\ell, r_\ell, x_{\ell+1})$ in Replay Buffer $R$
12:     **for** $s \leftarrow 1, \dots, S$ **do** ▷ CAQL Training; $S = 20$ by default
13:         Sample a random minibatch $B$ of $|B|$ transitions $\{(x_i, a_i, r_i, x_i')\}_{i=1}^{|B|}$ from $R$
14:         Initialize the refined batches $B'_{\text{df}} \leftarrow B$ and $B'_{\text{c}} \leftarrow B$;

   (i) IF DF is True:  $B'_{\text{df}} \leftarrow \{(x, a, r, x') \in B : \widetilde{q}_{x', \theta^{\text{target}}} > (Q_\theta(x, a) - r)/\gamma\}$

   (ii) IF C is True:  $B'_{\text{c}} \leftarrow \{(x, a, x', r) \in B : x' \in C_t(b)\}$

15:         For each $(x_i, a_i, r_i, x_i') \in B'_{\text{df}} \cap B'_{\text{c}}$, compute optimal action $a_i'$ using OPT(DTol):

$$a_i' \in \arg\max_{a'} Q_\theta(x_i', a')$$

16:         and the corresponding TD targets:

$$q_i = r_i + \gamma Q_{\theta^{\text{target}}}(x_i', a_i')$$

17:         For each $(x_i, a_i, r_i, x_i') \in B \setminus (B'_{\text{c}} \cap B'_{\text{df}})$, compute the approximate TD target:

   (i) IF C is True:  $\forall (x_i, a_i, r_i, x_i') \in B'_{\text{df}} \setminus B'_{\text{c}}, \quad q_i \leftarrow r_i + \gamma \hat{q}_{x_i', \theta^{\text{target}}}$

       where $\hat{q}_{x_i', \theta^{\text{target}}} = q^*_{c_i', \theta^{\text{target}}} + \langle \nabla_{x'} Q_{\theta^{\text{target}}}(x', a')|_{x'=c', a'=a^*_{c_i'}}, (x_i' - c_i') \rangle$,

       and $c_i' \in C_t(b)$ is the closest centroid to $x_i'$

   (ii) IF DF is True:  $\forall (x_i, a_i, r_i, x_i') \in B \setminus B'_{\text{df}}, \quad q_i \leftarrow r_i + \gamma \widetilde{q}_{x_i', \theta^{\text{target}}}$

18:         Update the Q-function parameters:

$$\theta \leftarrow \arg\min_\theta \frac{1}{|B|} \sum_{i=1}^{|B|} (Q_\theta(x_i, a_i) - q_i)^2$$

19:         Update the action function parameters:

$$w \leftarrow \arg\min_w \frac{1}{|B|} \sum_{i=1}^{|B|} (Q_\theta(x_i', a_i') - Q_\theta(x_i', \pi_w(x_i')))^2$$

20:         Update the target Q-function parameters:

$$\theta^{\text{target}} \leftarrow \tau\theta + (1 - \tau)\theta^{\text{target}}$$

21:     Decay the Gaussian noise:

$$\sigma \leftarrow \lambda\sigma, \lambda \in [0, 1]$$

---

## C DETAILS OF DUAL FILTERING

Recall that the Q-function NN has a nonlinear activation function, which can be viewed as a non-linear equality constraint, according to the formulation in (3). To tackle this constraint, Wong & Kolter (2017) proposed a convex relaxation of the ReLU non-linearity. Specifically, first, they assume that for given $x' \in X$ and $a' \in B_\infty(\overline{a})$ such that $z_1 = (x', a')$, there exists a collection of component-wise bounds $(l_j, u_j), j = 2, \dots, K-1$ such that $l_j \leq \hat{z}_j \leq u_j$. As long as the bounds are redundant, adding these constraints into primal problem $q_{x'}^*$ does not affect the optimal value. Second, the ReLU non-linear equality constraint is relaxed using a convex outer-approximation. In particular, for a scalar input $a$ within the real interval $[l, u]$, the exact ReLU non-linearity acting on $a$ is captured by the set

$$\mathcal{H}(l, u) := \{(h, k) \in \mathbb{R}^2 : h \in [l, u], k = [h]_+\}.$$

Its convex outer-approximation is given by:

$$\tilde{\mathcal{H}}(l, u) := \{(h, k) \in \mathbb{R}^2 : k \geq h, k \geq 0, -uh + (u-l)k \leq -ul\}. \tag{9}$$

Analogously to (3), define the *relaxed* NN equations as:

$$z_1 = (x', a'), \; a' \in B_\infty(\overline{a}, \Delta) \tag{10a}$$

$$\hat{z}_j = W_{j-1} z_{j-1} + b_{j-1}, \quad j = 2, \dots, K \tag{10b}$$

$$(\hat{z}_j, z_j) \in \tilde{\mathcal{H}}(l_j, u_j), \quad j = 2, \dots, K-1, \tag{10c}$$

where the third equation above is understood to be component-wise across layer $j$ for each $j \in \{2, \dots, K-1\}$, i.e.,

$$(\hat{z}_j(s), z_j(s)) \in \tilde{\mathcal{H}}(l_j(s), u_j(s)), \quad s = 1, \dots, n_j,$$

where $n_j$ is the dimension of hidden layer $j$. Using the relaxed NN equations, we now propose the following relaxed (convex) verification problem:

$$\tilde{q}_{x'}^* := \max_{\hat{z}, z} \quad c^\top \hat{z}_K + \delta_{B_\infty(\overline{a})}(a') + \sum_{j=2}^{K-1} \delta_{\tilde{\mathcal{H}}(l_j, u_j)}(\hat{z}_j, z_j) \tag{11a}$$

$$\text{s.t.} \quad \hat{z}_j = W_{j-1} z_{j-1} + b_{j-1}, \quad j = 2, \dots, K, \tag{11b}$$

where $\delta_\Lambda(\cdot)$ is the indicator function for set $\Lambda$ (i.e., $\delta_\Lambda(x) = 0$ if $x \in \Lambda$ and $\infty$ otherwise). Note that the indicator for the vector-ReLU in cost function above is understood to be component-wise, i.e.,

$$\delta_{\tilde{\mathcal{H}}(l_j, u_j)}(\hat{z}_j, z_j) = \sum_{s=1}^{n_j} \delta_{\tilde{\mathcal{H}}(l_j(s), u_j(s))}(\hat{z}_j(s), z_j(s)), \quad j = 2, \dots, K-1.$$

The optimal value of the relaxed problem, i.e., $\tilde{q}_{x'}^*$ is an upper bound on the optimal value for original problem $q_{x_i'}^*$. Thus, the certification one can obtain is the following: if $\tilde{q}_{x'}^* \leq (Q_\theta(x, a) - r)/\gamma$, then the sample $(x, a, x')$ is discarded for inner maximization. However, if $\tilde{q}_{x'}^* > (Q_\theta(x, a) - r)/\gamma$, the sample $(x, a, x')$ may or may not have any contribution to the TD-error in the hinge loss function.

To further speed up the computation of the verification problem, by looking into the dual variables of problem (11), in the next section we propose a numerically efficient technique to estimate a sub-optimal, upper-bound estimate to $\tilde{q}_{x'}^*$, namely $\tilde{q}_{x'}$. Therefore, one verification criterion on whether a sample drawn from replay buffer should be discarded for inner-maximization is check whether the following inequality holds:

$$\tilde{q}_{x'} \leq \frac{Q_\theta(x, a) - r}{\gamma}. \tag{12}$$

### C.1 SUB-OPTIMAL SOLUTION TO THE RELAXED PROBLEM

In this section, we detail the sub-optimal lower bound solution to the relaxed problem in (11) as proposed in Wong & Kolter (2017). Let $\nu_j, j = 2, \dots, K$ denote the dual variables for the linear equality constraints in problem (11). The Lagrangian for the relaxed problem in (11) is given by:

$$L(\hat{z}, z, \nu) = \left(c^\top \hat{z}_K + \nu_K^\top \hat{z}_K\right) + \left(\delta_{B_\infty(\overline{a})}(a') - \nu_2^\top W_1 z_1\right) +$$

$$\sum_{j=2}^{K-1} \left(\delta_{\tilde{\mathcal{H}}(l_j, u_j)}(\hat{z}_j, z_j) + \nu_j^\top \hat{z}_j - \nu_{j+1}^\top W_j z_j\right) - \sum_{j=1}^{K-1} \nu_{j+1}^\top b_j.$$

Define $\hat{\nu}_j := W_j^\top \nu_{j+1}$ for $j = 1, \ldots, K-1$, and define $\hat{\nu}_1^{x'} = (W_1^{x'})^\top \nu_2$, $\hat{\nu}_1^{a'} = (W_1^{a'})^\top \nu_2$. Then, given the decoupled structure of $L$ in $(\hat{z}_j, z_j)$, minimizing $L$ w.r.t. $(\hat{z}, z)$ yields the following dual function:

$$g(\nu) = \begin{cases} -\delta_{B_\infty(\overline{a})}^\star(\hat{\nu}_1^{a'}) - (\hat{\nu}_1^{x'})^\top x' - \sum_{j=2}^{K-1} \delta_{\tilde{\mathcal{H}}(l_j, u_j)}^\star\left(\begin{bmatrix} -\nu_j \\ \hat{\nu}_j \end{bmatrix}\right) - \sum_{j=1}^{K-1} \nu_{i+1}^\top b_i & \text{if } \nu_K = -c \\ -\infty & \text{else.} \end{cases}$$

(13)

Recall that for a real vector space $X \subseteq \mathbb{R}^n$, let $X^*$ denote the dual space of $X$ with the standard pairing $\langle \cdot, \cdot \rangle : X \times X^* \to \mathbb{R}$. For a real-valued function $f : X \to \mathbb{R} \cup \{\infty, -\infty\}$, let $f^* : X^* \to \mathbb{R} \cup \{\infty, -\infty\}$ be its convex conjugate, defined as: $f^*(y) = -\inf_{x \in X}(f(x) - \langle y, x \rangle) = \sup_{x \in X}(\langle y, x \rangle - f(x))$, $\forall y \in X^*$. Therefore, the conjugate for the vector-ReLU indicator above takes the following component-wise structure:

$$\delta_{\tilde{\mathcal{H}}(l_j, u_j)}^\star\left(\begin{bmatrix} -\nu_j \\ \hat{\nu}_j \end{bmatrix}\right) = \sum_{s=1}^{n_j} \delta_{\tilde{\mathcal{H}}(l_j(s), u_j(s))}^\star(-\nu_j(s), \hat{\nu}_j(s)), \quad j = 2, \ldots, K-1.$$

(14)

Now, the convex conjugate of the set indicator function is given by the set support function. Thus,

$$\delta_{B_\infty(\overline{a})}^\star(\hat{\nu}_1^{a'}) = \overline{a}^\top \hat{\nu}_1^{a'} + \Delta \|\hat{\nu}_1^{a'}\|_q,$$

where $\| \cdot \|_q$ is the $l_p$-dual norm defined by the identity $1/p + 1/q = 1$. To compute the convex conjugate for the ReLU relaxation, we analyze the scalar definition as provided in (9). Specifically, we characterize $\delta_{\tilde{\mathcal{H}}(l,u)}^\star(p, q)$ defined by the scalar bounds $(l, u)$, for the dual vector $(p, q) \in \mathbb{R}^2$. There exist 3 possible cases:

*Case I*: $l < u \le 0$:

$$\delta_{\tilde{\mathcal{H}}(l,u)}^\star(p, q) = \begin{cases} p \cdot u & \text{if } p > 0 \\ p \cdot l & \text{if } p < 0 \\ 0 & \text{if } p = 0. \end{cases}$$

(15)

*Case II*: $0 \le l < u$:

$$\delta_{\tilde{\mathcal{H}}(l,u)}^\star(p, q) = \begin{cases} (p+q) \cdot u & \text{if } p + q > 0 \\ (p+q) \cdot l & \text{if } p + q < 0 \\ 0 & \text{if } p = -q. \end{cases}$$

(16)

*Case III*: $l < 0 < u$: For this case, the sup will occur either on the line $-ux + (u-l)y = -ul$ or at the origin. Thus,

$$\begin{aligned}
\delta_{\tilde{\mathcal{H}}(l,u)}^\star(p, q) &= \left[\sup_{h \in [l,u]} p \cdot h + \frac{u}{u-l} \cdot (h-l) \cdot q\right]_+ \\
&= \left[\sup_{h \in [l,u]} \left(p + q \cdot \frac{u}{u-l}\right) \cdot h - q \cdot l \cdot \frac{u}{u-l}\right]_+ \\
&= \begin{cases} [(p+q) \cdot u]_+ & \text{if } \left(p + q \cdot \frac{u}{u-l}\right) > 0 \\ [p \cdot l]_+ & \text{if } \left(p + q \cdot \frac{u}{u-l}\right) < 0 \\ \left[-q \cdot l \cdot \frac{u}{u-l}\right]_+ = [p \cdot l]_+ & \text{if } \left(p + q \cdot \frac{u}{u-l}\right) = 0. \end{cases}
\end{aligned}$$

(17)

Applying these in context of equation (14), we calculate the Lagrange multipliers by considering the following cases.

*Case I*: $l_j(s) < u_j(s) \le 0$: In this case, since $z_j(s) = 0$ regardless of the value of $\hat{z}_j(s)$, one can simply remove these variables from problems (10) and (11) by eliminating the $j^{\text{th}}$ row of $W_{j-1}$ and $b_{j-1}$ and the $j^{\text{th}}$ column of $W_i$. Equivalently, from (15), one can remove their contribution in (13) by setting $\nu_j(s) = 0$.

*Case II*: $0 \le l_j(s) < u_j(s)$: In this case, the ReLU non-linearity for $(\hat{z}_j(s), z_j(s))$ in problems (10) and (11) may be replaced with the convex linear equality constraint $z_j(s) = \hat{z}_j(s)$ with associated dual variable $\mu$. Within the Lagrangian, this would result in a modification of the term

$$\delta_{\tilde{\mathcal{H}}(l_j(s), u_j(s))}(\hat{z}_j(s), z_j(s)) + \nu_j(s)\hat{z}_j(s) - \hat{\nu}_j(s)z_j(s)$$

to

$$\mu(z_j(s) - \hat{z}_j(s)) + \nu_j(s)\hat{z}_j(s) - \hat{\nu}_j(s)z_j(s).$$

Minimizing this over $(\hat{z}_j(s), z_j(s))$, a non-trivial lower bound (i.e., 0) is obtained only if $\nu_j(s) = \hat{\nu}_j(s) = \mu$. Equivalently, from (16), we set $\nu_j(s) = \hat{\nu}_j(s)$.

*Case III*: For the non-trivial third case, where $l_j(s) < 0 < u_j(s)$, notice that due to $\hat{\nu}$, the dual function $g(\nu)$ is not decoupled across the layers. In order to get a sub-optimal, but analytical solution to the dual optimization, we will optimize each term within the first sum in (13) independently. To do this, notice that the quantity in sub-case I in (17) is strictly greater than the other two sub-cases. Thus, the best bound is obtained by using the third sub-case, which corresponds to setting:

$$\nu_j(s) = \hat{\nu}_j(s)\frac{u_j(s)}{u_j(s) - l_j(s)}.$$

Combining all the previous analysis, we now calculate the dual of the solution to problem (11). Let $\mathcal{I}_j^- := \{s \in \mathbb{N}_{n_j} : u_j(s) \le 0\}$, $\mathcal{I}_j^+ := \{s \in \mathbb{N}_{n_j} : l_j(s) \ge 0\}$, and $\mathcal{I}_j := \mathbb{N}_{n_j} \setminus (\mathcal{I}_j^- \cup \mathcal{I}_j^+)$. Using the above case studies, a sub-optimal (upper-bound) dual solution to the primal solution $\tilde{J}_{x'}$ in problem (11) is given by

$$\tilde{q}_{x'} := -(\hat{\nu}_1^{a'})^\top \bar{a} - \Delta\|\hat{\nu}_1^{a'}\|_q - (\hat{\nu}_1^x)^\top x + \sum_{j=2}^{K-1}\sum_{s \in \mathcal{I}_j} l_j(s)[\nu_j(s)]_+ - \sum_{j=1}^{K-1} \nu_{j+1}^\top b_i, \qquad (18)$$

where $\nu$ is defined by the following recursion, termed the "dual" network:

$$\nu_K := -c, \quad \hat{\nu}_j := W_j^\top \nu_{j+1}, \quad j = 1, \dots, K-1, \quad \nu_j := D_j\hat{\nu}_j, \quad j = 2, \dots, K-1, \qquad (19)$$

and $D_j$ is a diagonal matrix with

$$[D_j](s,s) = \begin{cases} 0 & \text{if } s \in \mathcal{I}_j^- \\ 1 & \text{if } s \in \mathcal{I}_j^+ \\ \dfrac{u_j(s)}{u_j(s) - l_j(s)} & \text{if } s \in \mathcal{I}_j. \end{cases} \qquad (20)$$

## C.2 COMPUTING PRE-ACTIVATION BOUNDS

For $k \in \{3, \dots, K-1\}$, define the $k-partial$ NN as the set of equations:

$$z_1 = (x', a'), \quad \hat{z}_j = W_{j-1}z_{j-1} + b_{j-1}, \quad j = 2, \dots, k, \quad z_j = h(\hat{z}_j), \quad j = 2, \dots, k-1. \qquad (21)$$

Finding the lower bound $l_k$ for $\hat{z}_k$ involves solving the following problem:

$$\min_{\hat{z}, z} \quad e_s^\top \hat{z}_k \qquad (22)$$

$$\text{s.t.} \quad z_1 = (x', a'), \ a' \in B_\infty(\bar{a}, \Delta), \ \text{eq. (21)}, \qquad (23)$$

where $e_s$ is a one-hot vector with the non-zero element in the $s$-th entry, for $s \in \{1, \dots, n_k\}$. Similarly, we obtain $u_k$ by maximizing the objective above. Assuming we are given bounds $\{l_j, u_j\}_{j=2}^{k-1}$, we can employ the same convex relaxation technique and approximate dual solution as for the verification problem (since we are simply optimizing a linear function of the output of the first $k$ layers of the NN). Doing this recursively allows us to compute the bounds $\{l_j, u_j\}$ for $j = 3, \dots, K-1$. The recursion is given in Algorithm 1 in Wong & Kolter (2017) and is based on the matrix form of the recursion in (19), i.e., with $c$ replaced with $I$ and $-I$, so that the quantity in (18) is vector-valued.

## D  EXPERIMENTAL DETAILS

| Environment | State dimension | Action dimension | Action ranges |
|---|---|---|---|
| Pendulum | 3 | 1 | **[-2, 2]**, [-1, 1], [-0.66, 0.66] |
| Hopper | 11 | 3 | **[-1, 1]**, [-0.5, 0.5], [-0.25, 0.25] |
| Walker2D | 17 | 6 | **[-1, 1]**, [-0.5, 0.5], [-0.25, 0.25] |
| HalfCheetah | 17 | 6 | **[-1, 1]**, [-0.5, 0.5], [-0.25, 0.25] |
| Ant | 111 | 8 | **[-1.0, 1.0]**, [-0.5, 0.5], [-0.25, 0.25], [-0.1, 0.1] |
| Humanoid | 376 | 17 | **[-0.4, 0.4]**, [-0.25, 0.25], [-0.1, 0.1] |

Table 6: Benchmark Environments. Various action bounds are tested from the default one to smaller ones. The action range in bold is the default one. For high-dimensional environments such as Walker2D, HalfCheetah, Ant, and Humanoid, we only test on action ranges that are smaller than the default (in bold) due to the long computation time for MIP. A smaller action bound results in a MIP that solves faster.

| Hyper Parameters for CAQL and NAF | Value(s) |
|---|---|
| Discount factor | 0.99 |
| Exploration policy | $\mathcal{N}(0, \sigma = 1)$ |
| Exploration noise ($\sigma$) decay | 0.9995, 0.9999 |
| Exploration noise ($\sigma$) minimum | 0.01 |
| Soft target update rate ($\tau$) | 0.001 |
| Replay memory size | $10^5$ |
| Mini-batch size | 64 |
| Q-function learning rates | 0.001, 0.0005, 0.0002, 0.0001 |
| Action function learning rates (for CAQL only) | 0.001, 0.0005, 0.0002, 0.0001 |
| Tolerance decay (for dynamic tolerance) | 0.995, 0.999, 0.9995 |
| Lambda penalty (for CAQL with Hinge loss) | 0.1, 1.0, 10.0 |
| Neural network optimizer | Adam |

Table 7: Hyper parameters settings for CAQL(+ MIP, GA, CEM) and NAF. We sweep over the Q-function learning rates, action function learning rates, and exploration noise decays.

| Hyper Parameters for DDPG, TD3, SAC | Value(s) |
|---|---|
| Discount factor | 0.99 |
| Exploration policy (for DDPG and TD3) | $\mathcal{N}(0, \sigma = 1)$ |
| Exploration noise ($\sigma$) decay (for DDPG and TD3) | 0.9995 |
| Exploration noise ($\sigma$) minimum (for DDPG and TD3) | 0.01 |
| Temperature (for SAC) | 0.99995, 0.99999 |
| Soft target update rate ($\tau$) | 0.001 |
| Replay memory size | $10^5$ |
| Mini-batch size | 64 |
| Critic learning rates | 0.001, 0.0005, 0.0002, 0.0001 |
| Actor learning rates | 0.001, 0.0005, 0.0002, 0.0001 |
| Neural network optimizer | Adam |

Table 8: Hyper parameters settings for DDPG, TD3, and SAC. We sweep over the critic learning rates, actor learning rates, temperature,and exploration noise decays.

We use a two hidden layer neural network with ReLU activation (32 units in the first layer and 16 units in the second layer) for both the Q-function and the action function. The input layer for the Q-function is a concatenated vector of state representation and action variables. The Q-function has a single output unit (without ReLU). The input layer for the action function is only the state representation. The output layer for the action function has $d$ units (without ReLU), where $d$ is the action dimension of a benchmark environment. We use SCIP 6.0.0 (Gleixner et al., 2018) for the MIP solver. A time limit of 60 seconds and a optimality gap limit of $10^{-4}$ are used for all experiments. For GA and CEM, a maximum iterations of 20 and a convergence threshold of $10^{-6}$ are used for all experiments if not stated otherwise.

# E    ADDITIONAL EXPERIMENTAL RESULTS

## E.1    OPTIMIZER SCALABILITY

Table 9 shows the average elapsed time of various optimizers computing max-Q in the experiment setup described in Appendix D. MIP is more robust to action dimensions than GA and CEM. MIP latency depends on the state of neural network weights. It takes longer time with highly dense NN weights, but on the other hand, it can be substantially quicker with sparse NN weights. Figure 3 shows the average elapsed time of MIP over training steps for various benchmarks. We have observed that MIP is very slow in the beginning of the training phase but it quickly becomes faster. This trend is observed for most benchmarks except Humanoid. We speculate that the NN weights for the Q-function are dense in the beginning of the training phase, but it is gradually *structurized* (e.g, sparser weights) so that it becomes an easier problem for MIP.

| Env. [Action range] | GA | CEM | MIP |
|---|---|---|---|
| Hopper [-1.0, 1.0] | Med($\kappa$): 0.093, SD($\kappa$): 0.015 | Med($\kappa$): 0.375, SD($\kappa$): 0.059 | Med($\kappa$): 83.515, SD($\kappa$): 209.172 |
| Walker2D [-0.5, 0.5] | Med($\kappa$): 0.093, SD($\kappa$): 0.015 | Med($\kappa$): 0.406, SD($\kappa$): 0.075 | Med($\kappa$): 104.968, SD($\kappa$): 178.797 |
| HalfCheetah [-0.5, 0.5] | Med($\kappa$): 0.093, SD($\kappa$): 0.015 | Med($\kappa$): 0.296, SD($\kappa$): 0.072 | Med($\kappa$): 263.453, SD($\kappa$): 88.269 |
| Ant [-0.25, 0.25] | Med($\kappa$): 0.109, SD($\kappa$): 0.018 | Med($\kappa$): 0.343, SD($\kappa$): 0.054 | Med($\kappa$): 87.640, SD($\kappa$): 160.921 |
| Humanoid [-0.25, 0.25] | Med($\kappa$): 0.140, SD($\kappa$): 0.022 | Med($\kappa$): 0.640, SD($\kappa$): 0.113 | Med($\kappa$): 71.171, SD($\kappa$): 45.763 |

Table 9: The (median, standard deviation) for the average elapsed time $\kappa$ (in msec) of various solvers computing max-Q problem.

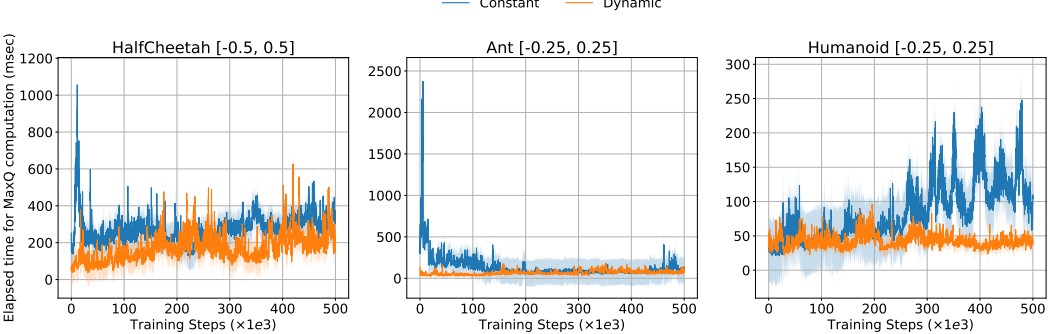

Figure 3: Average elapsed time (in msec) for MIP computing max-Q with constant and dynamic optimality gap.

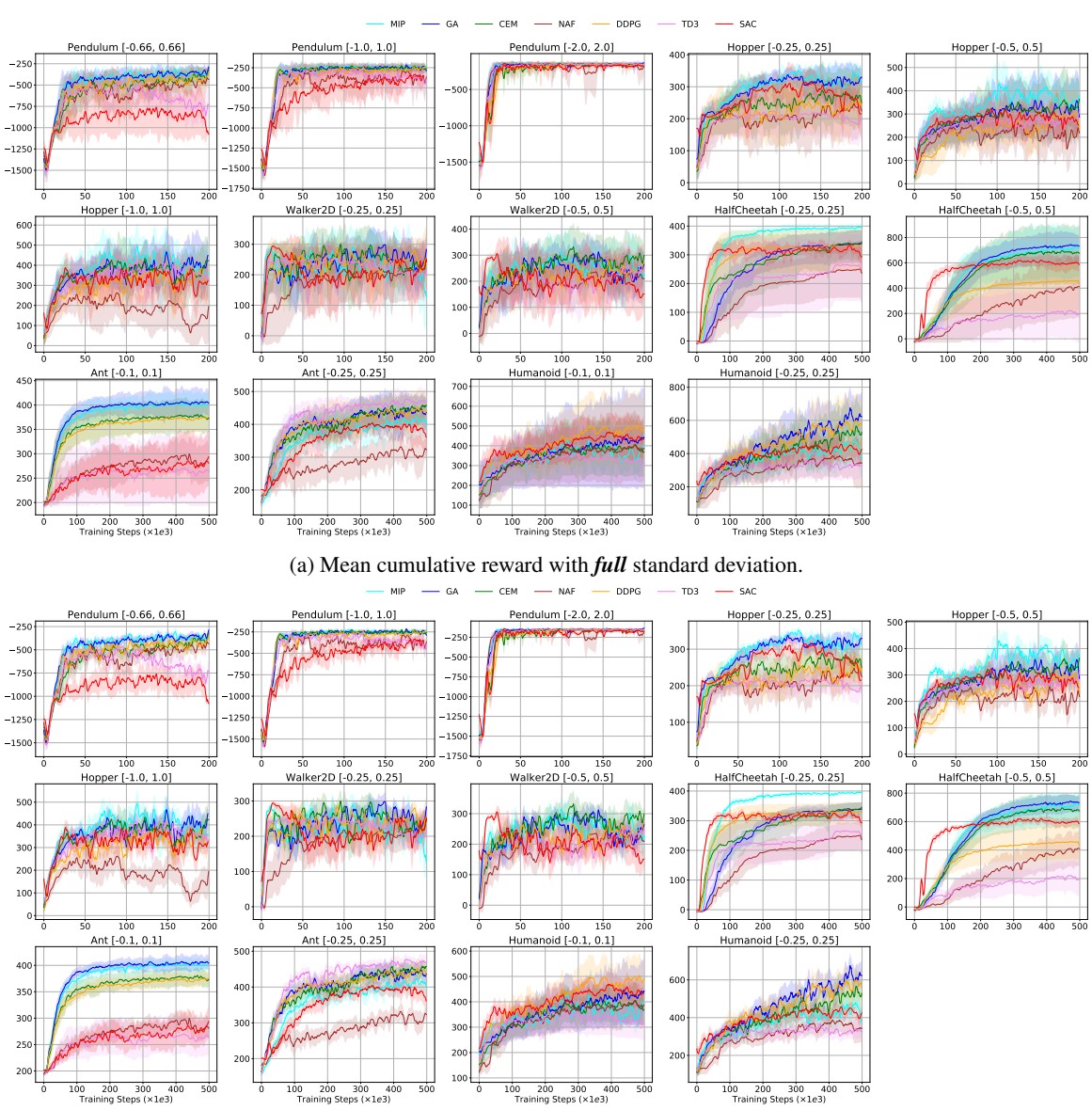

(a) Mean cumulative reward with **full** standard deviation.

(b) Mean cumulative reward with **half** standard deviation.

Figure 4: Mean cumulative reward of the best hyper parameter configuration over 10 random seeds. Shaded area is ± full or half standard deviation. Data points are average over a sliding window of size 6. The length of an episode is limited to 200 steps.

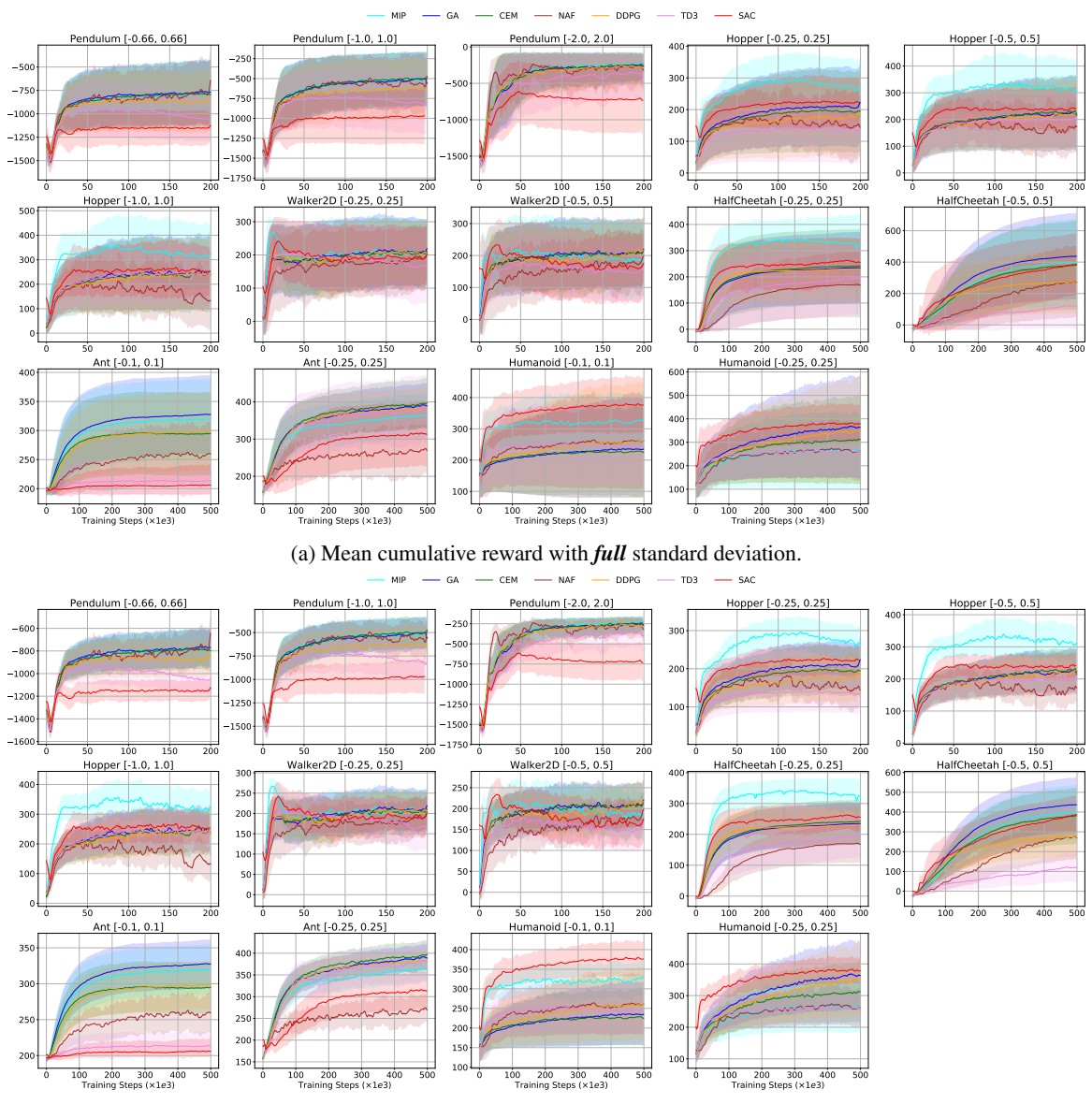

(a) Mean cumulative reward with *full* standard deviation.

(b) Mean cumulative reward with *half* standard deviation.

Figure 5: Mean cumulative reward over all 320 configurations (32 hyper parameter combinations × 10 random seeds). Shaded area is ± full or half standard deviation. Data points are average over a sliding window of size 6. The length of an episode is limited to 200 steps.

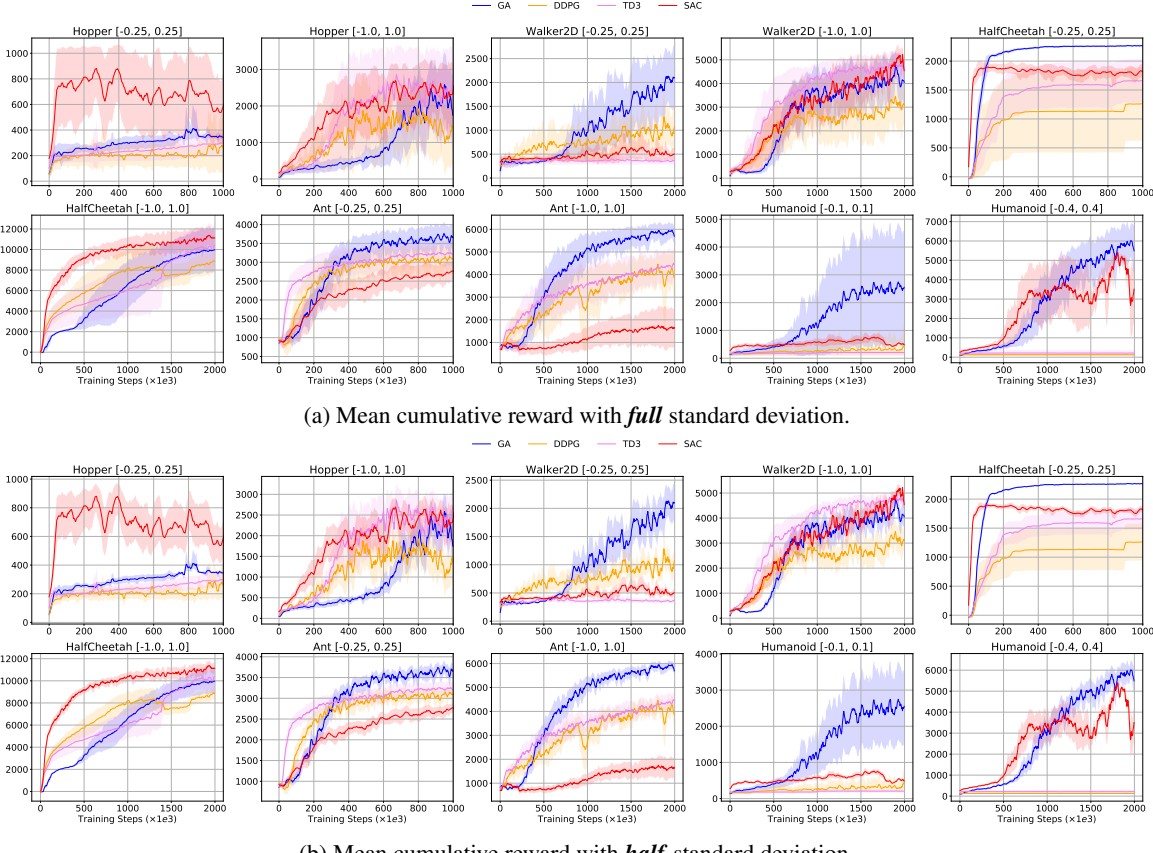

(a) Mean cumulative reward with **_full_** standard deviation.

(b) Mean cumulative reward with **_half_** standard deviation.

Figure 6: Mean cumulative reward of the best hyper parameter configuration over 10 random seeds. Shaded area is ± full or half standard deviation. Data points are average over a sliding window of size 6. The length of an episode is 1000 steps.

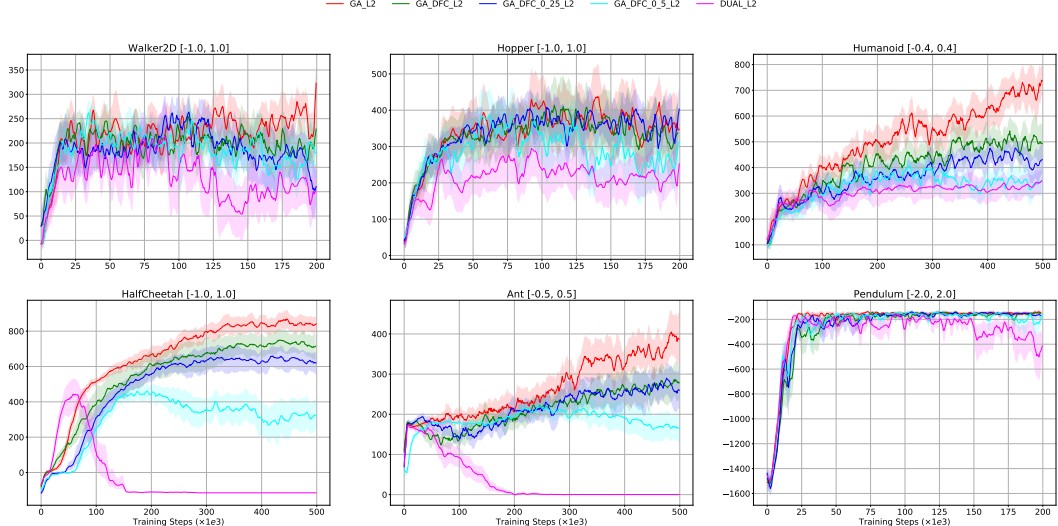

Figure 7: Mean cumulative reward of the best hyper parameter configuration over 10 random seeds. Shaded area is ± standard deviation. Data points are average over a sliding window of size 6. The length of an episode is limited to 200 steps.

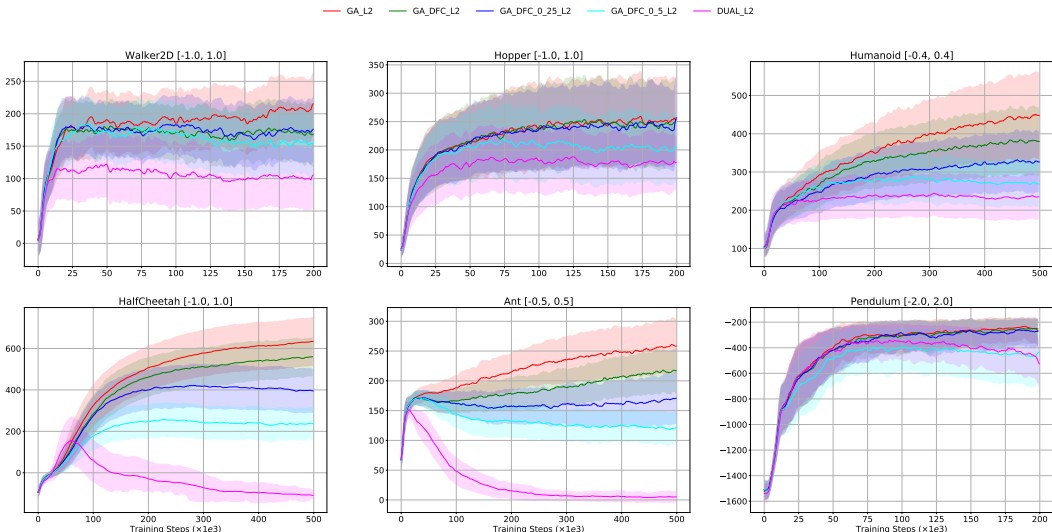

Figure 8: Mean cumulative reward over all 320 configurations (32 hyper parameter combinations × 10 random seeds). Shaded area is ± standard deviation. Data points are average over a sliding window of size 6. The length of an episode is limited to 200 steps.

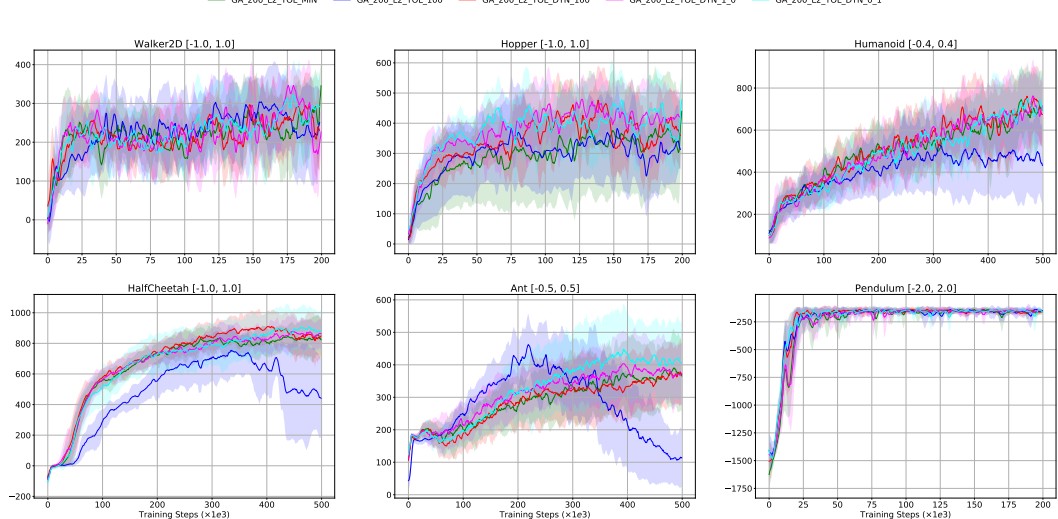

Figure 9: Mean cumulative reward of the best hyper parameter configuration over 10 random seeds. Shaded area is $\pm$ standard deviation. Data points are average over a sliding window of size 6. The length of an episode is limited to 200 steps.

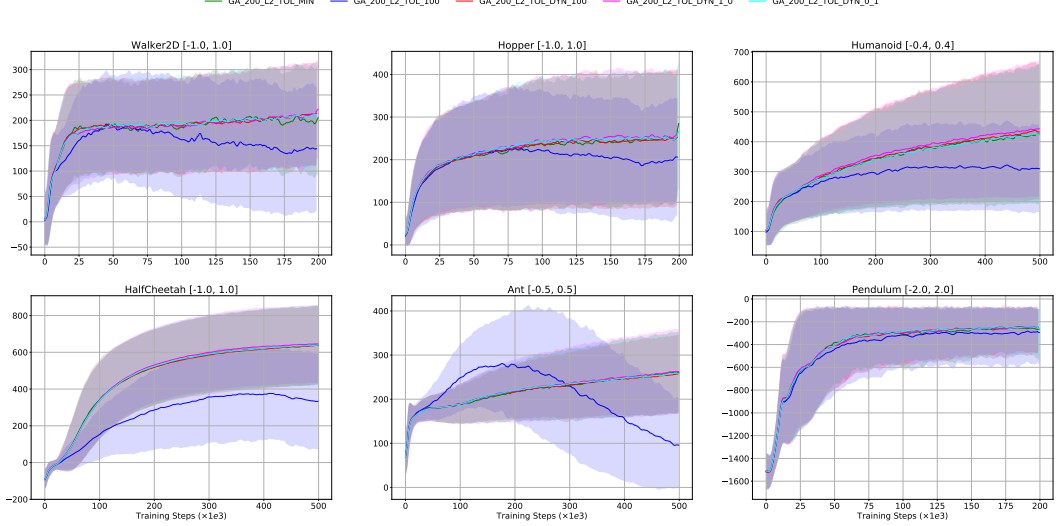

Figure 10: Mean cumulative reward over all 320 configurations (32 hyper parameter combinations $\times$ 10 random seeds). Shaded area is $\pm$ standard deviation. Data points are average over a sliding window of size 6. The length of an episode is limited to 200 steps.

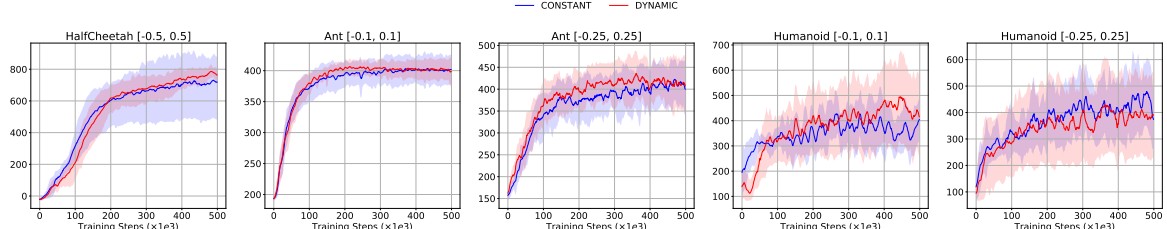

Figure 11: Comparison of CAQL-MIP with or without dynamic optimality gap on the mean return over 10 random seeds. Shaded area is $\pm$ standard deviation. Data points are average over a sliding window of size 6. The length of an episode is limited to 200 steps.

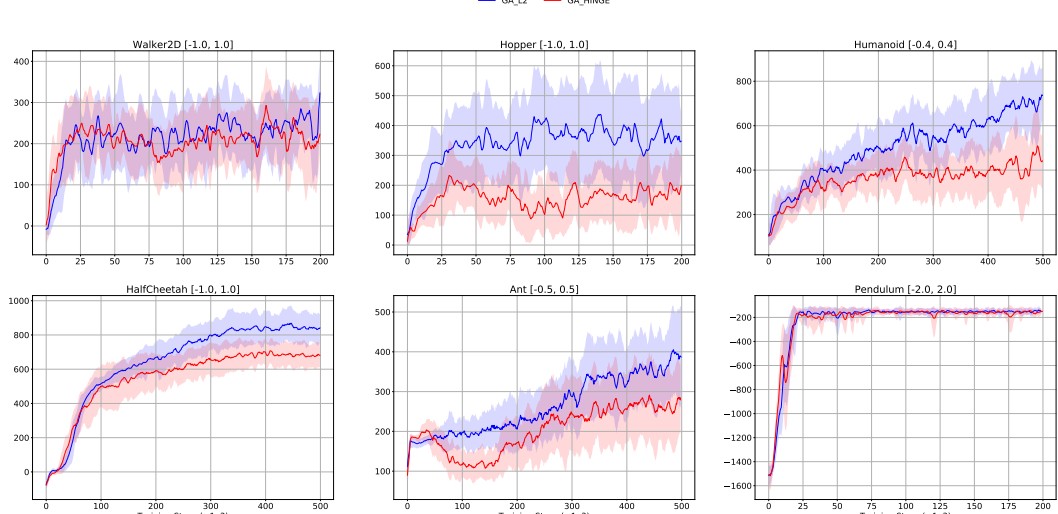

Figure 12: Mean cumulative reward of the best hyper parameter configuration over 10 random seeds. Shaded area is $\pm$ standard deviation. Data points are average over a sliding window of size 6. The length of an episode is limited to 200 steps.

| Env. [Action range] | $\ell_2$ CAQL-GA | Hinge CAQL-GA |
|---|---|---|
| Pendulum [-2, 2] | -244.488 $\pm$ 497.467 | **-232.307** $\pm$ 476.410 |
| Hopper [-1, 1] | **253.459** $\pm$ 158.578 | 130.846 $\pm$ 92.298 |
| Walker2D [-1, 1] | **207.916** $\pm$ 100.669 | 173.160 $\pm$ 106.138 |
| HalfCheetah [-1, 1] | **628.440** $\pm$ 234.809 | 532.160 $\pm$ 192.558 |
| Ant [-0.5, 0.5] | **256.804** $\pm$ 90.335 | 199.667 $\pm$ 84.116 |
| Humanoid [-0.4, 0.4] | **443.735** $\pm$ 223.856 | 297.097 $\pm$ 151.533 |

Table 10: The mean $\pm$ standard deviation of (95-percentile) final returns over all 320 configurations (32 hyper parameter combinations $\times$ 10 random seeds). The full training curves are given in Figure 12.

