# OpenReview forum: "CAQL: Continuous Action Q-Learning"
_ICLR.cc/2020/Conference — Accept (Poster)_

### Official Review · AnonReviewer2 · 2019-10-23
**Official Blind Review #2**

**Rating:** 6

**Review:**

The paper proposes a novel value-based continuous control algorithm by formulating the problem as mixed-integer programming. With this formulation, the optimal action (corresponding to the maximum action value) can be found by solving the optimization problem at each time step. To reduce the time complexity of the optimization, the author proposes several variants to approximately solve the problem. Results on robotics control are presented. The proposed looks interesting and could be useful in practice.

1. Section 4 of the paper can be improved. Although the author proposes several methods for approximating the optimal solution, it is unclear what message the author wants to convey. How to decide which approximation to use? Is there any situation where one of the approximations should be preferred?

2. In the experiments, the standard deviation is very large, so it is hard to claim the proposed method is better.

**Experience Assessment:**

I have published one or two papers in this area.

**Review Assessment: Checking Correctness Of Derivations And Theory:**

I did not assess the derivations or theory.

**Review Assessment: Checking Correctness Of Experiments:**

I assessed the sensibility of the experiments.

**Review Assessment: Thoroughness In Paper Reading:**

I read the paper at least twice and used my best judgement in assessing the paper.

---

> ### Author Response · Authors · 2019-11-10
> **Thanks for the helpful suggestions and questions on our paper. We provide brief responses to main issues/questions raised.**
>
> Which approximation to use?: In Section 4 we provide three techniques---dynamic tolerance, dual filtering, and clustering---to accelerate training (reduce computational cost). Dynamic tolerance is useful in a very general sense, and can be applied to any problem. It speeds up the MIP solver by adjusting its tolerance w.r.t. TD error, and in practice, improves the performance of CAQL significantly (see Section 5 for CAQL-GA and CAQL-MIP). We recommend using this method by default. Dual filtering and clustering both trade off training speed with performance (except in the case of hinge-Q learning, which has no loss of optimality when using dual filtering)---this is because one removes the action optimization step that corresponds to inactive and “less important” next states. These are the practical options (with tunable parameters) that allow users or developers to explore specific trade-offs suitable to specific application domains. We will elaborate on these points in the revised paper. You raise an important question of under which conditions (or in which environments) might these methods work better or worse. We do not have definitive answers and hope to address this question in future research.
>
> Large standard deviations: In general, model-free RL algorithms (including CAQL) indeed tend to have high variance during training. This also tends to obscure the significance of results in plots. Similar observations can be made of the results of other state-of-the-art methods such as DDPG and TD3. So to make the results more accessible and “visible” to readers, various tricks can be used. For example, in the TD3 paper (https://arxiv.org/pdf/1802.09477.pdf), the authors specify the shaded region of the training curves (see, e.g., their Figure 5) to cover just HALF of the standard deviation, while the shaded regions in all our plots cover the full standard deviation. We will explore ways to make performance differences more visible in our revised paper.
>
> That said, even with the current plots, we hope that it is evident that tasks with smaller action ranges (such as Hopper 0.25, HalfCheetah 0.25, Ant 0.1), CAQL-MIP has much lower variance in training than other methods (see Table 1 and the last paragraph on page 6 of the submission). This is a major benefit of using MIP-based solutions, despite the computational cost. For a clearer illustration, please find standalone graphs for CAQL-MIP at the following link: https://gofile.io/?c=Zno71b

---

### Official Review · AnonReviewer1 · 2019-10-23
**Official Blind Review #1**

**Rating:** 6

**Review:**

Summary:
This paper targets the maximization issue in continuous value based methods, especially Q Learning. The idea is to use Mixed Integer Programming (MIP) to solve the Q maximization step by formulating the neural network structure of the Q function as a constrained mixed integer program. Further improvements are made by approximating the MIP solution in order to make training/inference faster. The method is tested on tasks from the Mujoco domain and compared with other value based methods for continuous control. I found the paper simple to follow and well structured. The problem is well motivated too and the empirical analysis is quite rigorous.

The obvious concerns are regarding scalability of the method; both in terms of 1) using other forms of neural network components (ex. Other activation functions, Convolutional networks in the case of vision based problems such as robotic manipulation) and 2) problems that are inherently less sample efficient, i.e. cases where shortening the sampling horizon is not a feasible option for learning meaningful policies.

Overall, I feel the positive aspects more or less outweigh the drawbacks and therefore my vote is for a weak accept.


Comments/Questions:
- Table 8 description says hyper-parameter sweeps were done for temperature and exploration noise decay values but the table is missing their values.
- What happens when action range is increased from default? One of the reasons mentioned for constraining the action space is to validate how well policy-based methods work. To really take this point home, I feel it might be good to check with an increased action range.
- Controlling w.r.t symmetry of the env (esp. in Mujoco domains), thus reducing the number of actions by half, might help faster MIP computation times.
- Figure 3 x-axis runs till different values, but the description says training steps are 1000. How long are the experiments run for?
- Can the authors elaborate on why the episode length is decreased from 1000 to 200?


**Experience Assessment:**

I have read many papers in this area.

**Review Assessment: Checking Correctness Of Derivations And Theory:**

N/A

**Review Assessment: Checking Correctness Of Experiments:**

I carefully checked the experiments.

**Review Assessment: Thoroughness In Paper Reading:**

N/A

---

> ### Author Response · Authors · 2019-11-10
> **Thanks for the helpful suggestions and questions on our paper. We provide brief responses to main issues/questions raised.**
>
> One high-level clarification: While the MIP formulation and the approximations we develop are important (and probably the most novel element of the paper), the general CAQL framework, which supports other optimizers (e.g., GA, CEM) is also an contribution that can serve to investigate alternative approaches to continuous-action Q-learning. We leave the choice of an optimizer and speed-up methods to users, and let them decide based on the specific trade-offs in their applications.
>
> Other network types: To simplify notation, we focus our presentation of fully-connected feedforward network representations of Q-functions. That said, extending the model (including the MIP formulation) to convolutional networks (with ReLU activation and max pooling) is straightforward (see reference Anderson et al. 2019, at https://arxiv.org/abs/1811.01988). Handling additional activation functions is not problematic for GA or CEM; but we agree that it is certainly not straightforward with MIP, where some approximation would be required (e.g., sigmoids can be approximated in a piece-wise-linear fashion and encoded in a MIP). We leave this to future work.
>
> Hyper-parameter sweeps: Apologies and thanks for pointing this out. The values for the exploration decay hyper-parameter are given in Table 8, while there is no temperature parameter (the caption contains a typo). We will update the table/descriptions accordingly.
>
> Increased action range: This is an interesting suggestion! Our conjecture is that increasing the action ranges (to be above the default values) will not significantly change the results from the default (non-constrained) case because the action constraint is almost always inactive in these cases. But we will explore this suggestion experimentally to verify.
>
> Symmetry: This is an excellent suggestion and could certainly enhance the scalability of MIP-based optimization. While our primary intent is to provide benchmark results on “standard” MuJoCo experiments, we will extend our experiments using this idea. This should have an outsized impact on the MIP (vis-a-vis CEM, GA).
>
> Figure 3, Different values on x-axis, length of experiments: For the more difficult MuJoCo experiments (e.g., Ant, HalfCheetah, Humanoid), we set a longer training step of 500,000, while for simpler experiments (e.g., Pendulum, Hopper, Walker2D), we use a shorter training step of 200,000. In all the plots, performance is evaluated at every 1000 training steps.
>
> Why length-200 episodes?: We scoped down episode length in these experiments due to the heavy computational requirements of the MIP-based action solver. For problems with longer horizons, we require a larger network for better Q-function approximation, which in turn significantly increases the computation time needed by the MIP solver. Additionally, with a longer horizon, we need to train the model longer (more training steps). In order to keep the MIP solution time manageable (and to compare it to CEM, GA), we shortened the horizon length to 200 and parameterized the Q-function with a relatively simple 32x16 feed-forward network. Some of this motivation can be found on page 6 of the submission, but admittedly, it is a bit terse. We will justify the short-horizon setting in more detail in the revision.
>
> For a fairer comparison with state-of-the-art RL methods, we have also completed experimental results with the standard (horizon=1000) setup, comparing TD3, DDPG, and CAQL-GA (rather than CAQL MIP or CAQL-CEM). These results can be found at the following anonymous link: https://gofile.io/?c=x1iYJS  Similar to the setting in the TD3 paper (Fujimoto et al. 2018), we use a 400 x 300 feedforward network and observe a similar trend to that found with horizon 200.  In particular, with a constrained action range, CAQL-GA outperforms DDPG and TD3 for all Mujoco domains we test. For the default action range, CAQL-GA performs similar to TD3 in Hopper, Walker2D, HalfCheetah, while it outperforms the state-of-the-art in Ant and Humanoid by a significant margin.

---

### Public Comment · ~Olivier_Sigaud1 · 2019-10-07
**Missing reference to relevant papers**

Continuous Action Q-Learning is the title of a 2002 journal paper:

@article{millan2002continuous,
  title={Continuous-action Q-learning},
  author={Mill{\'a}n, Jos{\'e} Del R and Posenato, Daniele and Dedieu, Eric},
  journal={Machine Learning},
  volume={49},
  number={2-3},
  pages={247--265},
  year={2002},
  publisher={Springer}
}

There is also

@inproceedings{gaskett1999q,
  title={Q-learning in continuous state and action spaces},
  author={Gaskett, Chris and Wettergreen, David and Zelinsky, Alexander},
  booktitle={Australasian Joint Conference on Artificial Intelligence},
  pages={417--428},
  year={1999},
  organization={Springer}
}

which directly points to Lemon Baird's wire fitting.

Of course, these are not deep RL papers, but I think reading and properly citing these seminal papers is a good practice, as research in the domain did not start last year ;)

---

> ### Author Response · Authors · 2019-10-07
> **Will cite the papers in revision.**
>
> Apologies for not citing these (and other) earlier works, we will fix this in revision --- space constraints cut into our broader discussion of related work more than it should have.

---

### Public Comment · ~Haque_Ishfaq1 · 2023-12-12
**code for CAQL?**

Is there a publicly available code repo for CAQL?

---

> ### Author Response · Authors · 2023-12-13
> **CAQL opensource**
>
> https://github.com/google-research/google-research/tree/master/caql

---

### Decision · Program_Chairs · 2019-12-19

**Decision:**

Accept (Poster)

**Comment:**

All three reviewers gave scores of Weak Accept. AC has read the reviews and rebuttal and agrees that the paper makes a solid contribution and should be accepted.